

# Quantifying iceberg calving fluxes with underwater noise

Oskar Glowacki[1,2] and Grant B. Deane[1]

[1]Marine Physical Laboratory, Scripps Institution of Oceanography, La Jolla, USA
[2]Institute of Geophysics, Polish Academy of Sciences, Warsaw, Poland

*Correspondence to*: Oskar Glowacki (oglowacki@ucsd.edu)

**Abstract.** Accurate estimates of calving fluxes are essential to understand small-scale glacier dynamics and quantify the contribution of marine-terminating glaciers to both eustatic sea level rise and the freshwater budget of polar regions. Here we investigate the application of ambient noise oceanography to measure calving flux using the underwater sounds of iceberg-water impact. A combination of time-lapse photography and passive acoustics is used to determine the relationship between the mass and impact noise of 169 icebergs generated by subaerial calving events from Hans Glacier, Svalbard. The analysis
includes three major factors affecting the observed noise: 1. fluctuation of the thermohaline structure, 2. variability of the ocean depth along the waveguide, and 3. reflection of impact noise from the glacier terminus. A correlation of 0.76 is found between the (log-transformed) kinetic energy of the falling iceberg and the corresponding acoustic energy. An error-in-variables linear regression is applied to estimate the coefficients of this relationship. Energy conversion coefficients for non-transformed variables are $8 \times 10^{-7}$ and 0.92, respectively for the multiplication factor and exponent of the power law. As
we demonstrate, this simple model can be used to measure solid ice discharge from Hans Glacier. Uncertainty in the estimate is a function of the number of calving events observed; 50 % is expected for 8 blocks dropping to 20 % and 10 %, respectively, for 40 and 135 calving events. It may be possible to lower these errors if the influence of different calving styles on the received noise spectra can be determined.

## 1 Introduction

### 1.1 The role of iceberg calving in glacier retreat and sea-level rise

The contribution of glaciers and ice sheets to the eustatic sea-level rise (SLR) between 2003 and 2008 has been estimated to be $1.51 \pm 0.16$ mm of sea-level equivalent (SLE) per year (Gardner et al., 2013). Cryogenic freshwater sources were responsible for approximately $61 \pm 19$ % of the total SLR observed in the same period. Iceberg calving, defined as





mechanical loss of ice from the edges of glaciers and ice shelves (Benn et al., 2007), is thought to be one of the most important components of the total ice loss. As examples, solid ice discharge accounts for around 32 % to 40 % of the mass loss from the Greenland ice sheet (Enderlin et al., 2014; van den Broeke et al., 2016) and iceberg calving in Patagonia dominates glacial retreat (Schaefer et al., 2015). The exact partitioning between ice mass loss caused by calving fluxes,

submarine melting and surface runoff changes geographically and needs to be measured separately at each location. Calving from tidewater glaciers is driven by different mechanisms, including buoyant instability, longitudinal stretching, and terminus under-cutting (Van Der Veen, 2002; Benn et al., 2007). The latter results from submarine melting and is often considered to be a major trigger of ice breakup at the glacier front (Bartholomaus et al., 2013; O'Leary and Christoffersen, 2013). In support of this idea, the solid ice discharge from tidewater glaciers was found to be highly correlated with ocean

temperatures (Pętlicki et al., 2015; Luckman et al., 2015; Holmes et al., 2019), which are expected to increase significantly as a result of climate shifts (IPCC, 2013). Thus, accurate estimates of calving fluxes from marine-terminating glaciers are crucial to both understanding glacier dynamics and predicting their future contribution to SLR and the freshwater budget of the polar seas. Obtaining these estimates require remote sensing techniques, which enable the observation of dynamic glacial processes from a safe distance.

Satellite imagery is an effective way to study large-scale, relatively slow changes at the ice-ocean interface, such as the disintegration of the 15 km long ice tongue from Jakobshavn Isbræ in 2003 in Greenland (Joughin et al., 2004) or the spectacular detachment of 32,000 km$^2$ of glacial ice from Larsen B Ice Shelf in 2002 in Antarctica (Scambos et al., 2003, 2004; Rignot et al., 2004). For fast-flowing ice masses, changes of terminus position caused by both solid ice discharge and

glacier flow must be clearly separated. Consequently, satellite imagery is more limited for observing calving events, which typically occur on sub-diurnal time scales and are often not greater than 1,000 m$^3$ in volume for most tidewater glaciers in Svalbard or Alaska (e.g. Chapuis and Tetzlaff, 2014). Moreover, weather conditions in polar regions often manifest themselves with a thick layer of clouds, fog and precipitation in the form of snow or rain making it difficult to track iceberg calving continuously using optical techniques, such as surface photography or terrestrial laser scanning. These difficulties

provide the motivation for investigating the use of underwater noise to quantify calving fluxes.

## 1.2 Measuring ice discharge – tools and methods

Many different methods have been developed to measure ice discharge from marine-terminating glaciers. Passive glacier seismology, also called 'cryoseismology' (Podolskiy and Walter, 2016), is probably one of the most mature, widespread and

useful tools; broadband seismometers have been widely installed in remote areas near calving glaciers since the pioneering work of Qamar and St. Lawrence (1983). Seismic signals associated with subaerial calving originate from two main mechanisms: 1. the free-fall of ice blocks onto the sea surface (Bartholomaus et al., 2012), and 2. interactions between detaching icebergs and their glacier terminus (e.g. Ekström et al., 2003; Murray et al., 2015). The latter interactions, also





known as 'glacial earthquakes', are caused by large, cubic-kilometer scale icebergs of full-glacier height, and the resulting seismic magnitude is not related to the iceberg volume in a simple manner (Sergeant et al., 2016). Low-frequency signals of glacial earthquakes provide unique dynamical constraints for mechanical calving models, but are significantly affected by local conditions, including glacier size and fjord geometry. Higher frequency (>1 Hz) calving seismicity from iceberg-ocean

interactions, constantly detected by distant seismic networks (e.g. O'Neel et al., 2010; Köhler et al., 2015), usually peaks between 1 and 10 Hz (Bartholomaus et al., 2015; Köhler et al., 2015). Both frequency content and amplitudes of high frequency signatures are found to be independent of iceberg volumes (O'Neel and Pfeffer, 2007; Walter et al., 2012). Bartholomaus et al. (2015) applied generalized linear models to correlate various properties of seismic signals originating at Yahtse Glacier, Alaska with estimates of iceberg sizes divided into 7 classes. They identified icequake duration as the most

significant predictor of iceberg volume. Based on this study, Köhler et al. (2016; 2019) successfully reconstructed a record of total frontal ablation at Kronebreen, Svalbard using seismic data calibrated with satellite images and lidar volume measurements.

Recently, Minowa et al. (2018, 2019) demonstrated the potential of using surface waves generated by falling icebergs to

quantify calving flux. They found a strong correlation between calving volumes estimated from time-lapse camera images and the maximum amplitudes of the waves. Other methods for quantifying ice discharge from marine-terminating glaciers, including surface photography (e.g. How et al., 2019), terrestrial laser scanning (e.g. Pętlicki and Kinnard, 2016) or ground-based radar imaging (e.g. Chapuis et al., 2010), are usually used for short-term measurements.

## 20  1.3 Studying iceberg calving with underwater noise

The approach investigated here is an example of ambient noise oceanography, which extracts environmental information from the underwater noise field (Clay and Medwin, 1977). Ambient noise oceanography may offer some advantages over other, more well-developed methods for the study of the interactions between land-based ice and the ocean. Low-cost hydrophones are easily deployed from small boats in front of marine-terminating glaciers and acoustic data can be gathered

continuously for several months or longer with a high (> 10,000 Hz) sampling rate and low maintenance. Measurements are insensitive to lighting conditions such as fog, cloud coverage and the polar night, humidity and intensity of precipitation. Moreover, acoustic signals recorded in glacial bays and fjords also contain signatures of ice melt associated with impulsive bubble release events (Urick, 1971; Tegowski et al., 2011; Deane et al., 2014; Pettit et al., 2015; Glowacki et al., 2018). While currently no quantitative models exist to estimate melt rates from underwater noise, the potential idea to

simultaneously measure submarine melting and calving, two major processes acting at the glacier-ocean interface, is worth mentioning.





Quantifying iceberg calving by 'listening to glaciers' was first proposed by Schulz et al. (2008) who suggested long-term deployments of hydrophones (underwater microphones) and pressure gauges, in addition to more traditional measurements of water temperature and salinity, to study signals of ice discharge together with accompanying hydrographic and wave conditions. Following this novel idea, independent studies conducted in Svalbard (Tegowski et al., 2012) and Alaska (Pettit, 2012) showed the first waveforms and spectra of the sounds generated by impacting ice blocks. Pettit (2012) provided an explanation for individual components of the signal, including low-frequency onset, pre-calving activity, mid-frequency block impact, iceberg oscillations, and mini-tsunami and seiche action. Encouraged by these initial results, Glowacki et al. (2015) analyzed 10 subaerial and 2 submarine calving events identified on both acoustic recordings and time-lapse photography made in front of Hans Glacier, Svalbard. A spectral analysis of 3 different calving types, named 'typical subaerial', 'sliding subaerial', and 'submarine', showed that they radiated underwater noise in distinct spectral and temporal patterns but all with a spectral peak between 10 and 200 Hz. Most importantly, acoustic emission below 200 Hz was highly correlated with block impact energy in a simple model. The dimensionless coefficient converting impact energy to acoustic energy at the calving impact point was found to be $5.16 \times 10^{-10}$ and the power exponent was assumed to be 1. These results demonstrated the potential of ambient noise oceanography to quantify solid ice discharge from tidewater glaciers. However, this earlier analysis was limited by the small number of subaerial calving events analyzed (10), lack of a full error analysis, and the unrealistic assumption of simple cylindrical spreading of acoustic waves in the water column.

To address these issues, we conducted a new study covering a total number of 169 subaerial calving events observed with time-lapse photography at Hans Glacier, Svalbard. Impact energies generated by falling icebergs are estimated with error bars and related to received acoustic signals. The total noise energy resulting from block-water impact is calculated using a standard sound propagation model Bellhop (Porter, 2011), which requires bathymetry data and sound speed profiles as inputs. Variability in transmission losses associated with sound wave reflections from an idealized, flat glacier terminus is also accounted for. The analysis shows that impact energy is strongly correlated with acoustic emission below 100 Hz. We present a new energy conversion efficiency calculated with this more detailed physical model and demonstrate how cumulative values of kinetic energy and ice mass loss can be found by integrating impact noise over a specified number of subaerial calving events.

## 2 Study area

### 2.1 General setting

Hans Glacier is a retreating, grounded, polythermal tidewater glacier terminating in Hornsund Fjord, Svalbard (Fig. 1). It covers an area of around 54 km² and is more than 15 km long (Błaszczyk et al., 2013). The glacier has a 1.5 km-wide active





calving front with an average height of around 30 m (Błaszczyk et al., 2009). The mean thickness and total volume of Hans Glacier is estimated to be 171 m and $9.6 \pm 0.1$ km$^3$, respectively (Grabiec et al., 2012). The surface flow of the glacier is dominated by basal motion in the ablation area (Vieli et al., 2004) and the mean annual flow velocity near the terminus and its calving flux is estimated to be 150 m y$^{-1}$ and $38.1 \times 10^6$ m$^3$ y$^{-1}$, respectively (Błaszczyk et al., 2009). The average

retreat rate of the glacier during $2005 - 2010$, 44 m y$^{-1}$, was more than twice the rate observed between 1900 and 2010 (Grabiec et al., 2012). These characteristics are representative of Svalbard's tidewater glaciers making the bay of Hans Glacier a good study site.

Both glacial behavior and the propagation of sound are sensitive to temporal variability in thermohaline structure of water

masses in the bay (Pętlicki et al., 2015; Glowacki et al., 2016). The calving activity of Hans Glacier is largely controlled by melt-driven undercutting of the ice cliff, driven by thermodynamic processes and wave erosion (Pętlicki et al., 2015). The water temperature and salinity in the center of the bay ranged from $-1.8\,℃$ to more than $2.0\,℃$ and from 30 PSU to almost 35 PSU during 2015 and 2016 (Moskalik et al., 2018). Significant wave height observed in the study site reached a maximum value of around 1.5 m over the period of August – November 2015 (Herman et al., 2019). A geomorphological

map of the bay reveals complicated structures in the seabed created by dynamic glacial processes acting after the Little Ice Age, including terminal moraines, flat areas and iceberg-generated-pits, to name a few (Ćwiąkała et al., 2018). The water depth along a transect parallel to the glacier terminus ranges from less than 20 m to almost 90 m (see Fig. 5 in Moskalik et al., 2018).

**2.2 Calving activity and sound propagation conditions**

The main dataset consists of more than a thousand subaerial calving events observed between July 30 and September 15, 2016 with three time-lapse cameras and two acoustic buoys deployed in the glacial bay (Fig. 1). At least 20 ice blocks calved each day. It was not always possible to unambiguously identify a calving event in both the image and acoustic datasets; the occurrence of more than one iceberg detachment between the two consecutive images resulted in ambiguity in the acoustic

data. Moreover, dense fog, rain, or otherwise unfavorable lighting conditions would at times obscure the terminus. From the total calving inventory, a subset of $N = 169$ events that were unambiguously identified in both the images and acoustic recordings (Figs. 1 and 2) were analyzed.

Measurements of ocean temperature and salinity in the bay revealed upward-refracting sound speed profiles with velocities

changing from around 1440 m s$^{-1}$ just below the surface to almost 1470 m s$^{-1}$ close to the bottom (Fig. 2a-d). The sound speed gradient between the surface layer and deeper layers, which controls refraction and transmission loss, is driven by fresh meltwater and was clearly increasing during the study period. Moreover, significant differences in sound velocity





profiles taken on the same day were also observed between different locations perpendicular and parallel to the glacier terminus, driven by a complex and three-dimensional distribution of the thermohaline field in the bay. The ocean depth between the locations of calving events and the two acoustic buoys varied from 10 m on underwater sills at the buoy location to more than 80 m in the western part of the bay near the terminus (Fig. 3). The bathymetry profiles were very

different for the two buoy locations, with a more variable depth observed in the case of the buoy deployed further from the glacier cliff.

## 3 Methods and data analysis

The development of underwater acoustics as a new tool to quantify calving fluxes requires thorough understanding of the causal relationship between the energy of the ice-water interaction and the resulting noise emission. In this section we

discuss all steps that are necessary to complete this task. They are illustrated in Fig. 4 and described in detail in the following subsections. Firstly, a time-lapse camera is used to estimate iceberg dimensions and block impact energies (3.1). Secondly, an underwater noise from iceberg-water impact is recorded at a safe distance from the glacier terminus and analyzed to find its amplitude-frequency characteristics (3.2). Then, in order to calculate impact noise energy at source, two factors have to be considered: 1. transmission loss in a waveguide, which depends on the distance to the buoy, sea bottom properties along the

propagation path and variable hydrographic conditions (3.3 and 3.4.1), and 2. the potential contribution of acoustic energy reflected from the underwater part of the glacier terminus on the received calving noise (3.4.2). Finally, a simple model relating impact noise energy to the kinetic energy of the falling ice block is proposed (3.5). The parameters of this model are derived and applied to the available calving inventory to quantify ice mass loss integrated over a limited number of calving events observed at Hans Glacier, Svalbard (4).

## 3.1 Photographic observation of calving events

Images of Hans Glacier terminus were taken every 15 minutes from 3 locations ('Cam 1-3' in Fig. 1) continuously between July 30 and September 15, 2016 using Canon EOS 1100 D cameras (4272 x 2848 pixel resolution and 18 mm focal length). The three cameras were not perfectly synchronized, which in fact enabled better separation of individual iceberg calving

events occurring shortly after one other. Additionally, a GoPro Hero 3+ camera was placed closer to the terminus to take pictures of the narrow ice cliff segment ('GoPro' in Fig. 1). This camera took images at a much higher rate of 1 per second but was not always active during the deployment. Iceberg volume and drop height were estimated using images from Cam 1, which had the most perpendicular orientation to the glacier front of all the cameras. The irregular shape of the ice cliff provided registration features, which were identified in both Landsat-8 satellite images (with resolution of 15 m) and the

camera images, enabling a precise calibration of the camera geometry. With the camera geometry calibrated, it was possible to determine the locations of calving events within the bay.





Following Minowa et al. (2018), the volumes of the calved ice blocks are estimated from the area at the glacier terminus exposed by the calving event. Newly exposed area is identified from differences between a pair of images taken by Cam 1 (see section 1 in SI for details). Exposed area in pixels squared, $A_{\mathrm{img}}$, is converted to its real value in m², $A_c$, using the formula:

$$A_c = \frac{A_{\mathrm{img}} d^2}{F^2},$$ (1)

where $d$ is the distance between the camera and drop location and $F$ is the camera focal length. Guided by previous reports on iceberg dimensions observed in Svalbard (Dowdeswell and Forsberg, 1992), we assumed that the thickness of the calved iceberg is proportional to the square root of the exposed area. Then, the iceberg volume is given by:

$$V = C A_c^{3/2},$$ (2)

where $C$ is a constant scaling factor, which is reported to be around 0.12 (Åström et al., 2014; Pętlicki and Kinnard, 2016). The drop height, $h$, is measured as a vertical distance between the sea surface and the mid-point of the falling ice block, converted from pixels to meters. Finally, the kinetic energy of the impacting ice block, $E_{\mathrm{imp}}$, is given by:

$$E_{\mathrm{imp}} = Mgh = V\rho_i gh,$$ (3)

where $\rho_i$ is the ice density, set to be a constant 917 kg m⁻³, $g = 9.81$ m s⁻² is the acceleration due to gravity, and $M$ is the iceberg mass. Equation (3) for $E_{\mathrm{imp}}$ is based on the assumption that there is no energy dissipated during the free fall of an iceberg. In reality, energy is dissipated though various physical mechanisms, such as: friction between an ice block and glacier terminus, momentum transfer at the early stage of the water entry, drag during the immersion phase, and block disintegration, which can happen at different stages of calving. However, the details of these hydrodynamic processes lie beyond the scope of this work. Because they are not included, equation (3) for $E_{\mathrm{imp}}$ provides an upper bound of the total amount of energy available for noise production during the block-water interaction.

## 3.2 Impact noise recordings and analysis

The acoustic data were recorded continuously between July 30 and September 15, 2016 using two HTI-96-MIN omnidirectional hydrophones deployed at depths of 40 m and 22 m respectively in front of Hans Glacier ('A1' and 'A2' in



Fig. 1). The hydrophones have a sensitivity of $-164$ dB re 1 V µPa$^{-1}$ and were sampled at a rate of 32 kHz at a resolution of 16 bits. The horizontal distance between the moorings and locations of calving events ranged between 700 m - 1500 m for the closer buoy, and between 1800 m - 2100 m for more distant buoy.

The sound produced by calving events was identified manually, based on timing determined from the time-lapse cameras and deviations from median sound level at frequencies below 200 Hz (see section 2 in SI for details). Power spectral density estimates, $P_{xx}$, were calculated for each calving event using the Welch method with a 16,384-point fast Fourier transform and a 50 % segment overlap. The acoustic energy of the block-water impact at the buoy, $E_{ac, obs}$, was subsequently calculated by low-pass filtering the noise record at $f_{low}$ and then integrating the mean-square pressure, $p_{low}^2$, over the event duration:

$$E_{ac, obs} = \frac{4\pi}{\rho_w c} \int_{t_{start}}^{t_{end}} p_{low}^2 dt. \tag{4}$$

The sound speed, $c$, and water density, $\rho_w$, in (4) were set to 1450 m s$^{-1}$ and 1025 kg m$^{-3}$, respectively. The factor of $4\pi$ accounts for the surface area of a unit sphere, over which the noise signal must be integrated to obtain total noise energy in Joules. The selection of the cutoff frequency of the filter ($f_{low} = 100$ Hz) is discussed in section 4.2. The background noise
energy, $E_{ac, bckg}$, for each event was computed analogously using noise segments of the same length as the corresponding calving signal, recorded just before the ice block impact.

### 3.3 Hydrographic and bathymetric data

An overview of the temperature and salinity structure in the study site and its influence on the propagation of sound
throughout the bay has been provided by Glowacki et al. (2016). In this study, temperature and salinity profiles were taken on August 1, 12, 30 and September 9, 2016 with an SAIV SD208 CTD probe at 11 stations located on transects perpendicular and parallel to the glacier terminus (red and blue dashed lines, respectively, in Fig. 1). Sound velocity was calculated from the CTD data according to Chen and Milleros' formulae adopted by UNESCO (Chen and Millero, 1977).

Each calving event has its own, unique set of hydrographic and bathymetric data used for modelling sound propagation, determined in the following way. A closest sound speed profile was assigned to each calving event according to its time in groups of 32, 46, 61 and 30 detachments (see Fig. 2). Additional CTD casts were taken in 2017 after significant recession of the Hans Glacier. These profiles provided information on bottom depths in 11 additional positions located near the glacier terminus position from 2016, which are not covered by the bathymetry data (0.1 m resolution) collected during multibeam
surveys (Fig. 1). We selected 5 bathymetry profiles, separately for two acoustic buoys, that lie along a straight line between




the mooring location and CTD stations belonging to the transect that is closest to the ice cliff. Ocean depths in these sections were then interpolated into a 1-m grid using shape-preserving, piecewise cubic interpolation (Fritsch and Carlson, 1980; Fig. 3). Despite the fact that a high level of variability in the thermohaline structure is expected and there is a lack of detailed bathymetry data close to the glacier terminus, the uniquely assigned sound speed profile and interpolated bathymetry are the
best available approximation of real conditions prevailing during the study period.

### 3.4 Attenuation of the calving noise in a glacial bay

### 3.4.1 Noise transmission loss

The underwater sound of a calving event must travel through the water column before reception at an acoustic buoy,
typically several 10's of water depths in range or more. Along its path, the signal undergoes multiple reflections from the sea surface and the sea floor and refracts because of changes in sound speed caused by the spatial and temporal variability of the thermohaline structure. These processes result in significant loss of the total signal energy and change the frequency spectrum of the noise observed at the receiver. These effects must be carefully modeled before the calving signature can be quantified in terms of ice block impact energy.

Here we used the standard ray propagation model Bellhop to compute transmission losses, $TLs$ (Porter, 2011). The number of beams was set to 2000, with launching angles ranging from -80 to 80 ° in respect to the sea surface. Guided by previous geomorphological studies (Görlich, 1986; Staszek and Moskalik, 2015), we assumed that the dominant sediment type in the study area is a clayey silt; density, sound speed and attenuation were taken to be $1.4\,\mathrm{g\,cm^{-3}}$, $1530\,\mathrm{m\,s^{-1}}$, and $0.1\,\mathrm{dB\,m^{-1}}$
$\mathrm{kHz^{-1}}$ respectively (Hamilton, 1970, 1976). Smoothing bathymetry and sound velocity profiles is highly recommended when using Bellhop to predict acoustic energy levels (Porter, 2011). The bathymetry profile for a selected calving event was spatially smoothed with a moving boxcar filter with a window size of $20\lambda$, where $\lambda = cf^{-1}$ is the wavelength of sound at the frequency of interest. The median sound speed profile calculated from the set of profiles measured at the closest time to the event occurrence was also spatially smoothed with moving average over 5 m. A baseline (most probable) transmission loss
was computed using the environmental data described above, assuming a source frequency of 50 Hz, which corresponds to the peak in the source spectrum, and a realistic source depth of 5 m.

The longest dimension of the calving icebergs is comparable to or greater than a wavelength over the impact noise frequencies, and all points distributed along the ice edge and its close vicinity are considered here to be incoherent noise
sources. Accordingly, the incoherent mode of propagation in Bellhop was used to compute $TL$. Finally, to investigate possible variability in $TL$, the simulations were repeated at 100 Hz with the bathymetry-smoothing window changed to $10\lambda$, ocean depth set to the water median depth and the sound speed profile taken to be each of the median profiles in turn.



### 3.4.2 Contribution from terminus-reflected noise

The Bellhop model does not easily account for sound reflected from the underwater part of the glacier terminus, which is potentially an important component of the total acoustic energy received at the buoy. The effect of the glacier terminus on observed calving noise, which may be quite complicated, is considered here.

Figure S3 in Supplementary Information illustrates the direct reflection of sound by the terminus, which is one possible propagation path but there are more, such as a surface or bottom reflection followed by reflection by the terminus, and so on. All possible paths can be enumerated using a series of image sequences (Deane and Buckingham, 1993) and could, in principle, be investigated. However, we have simplified the problem by considering only energy reflected directly by the terminus, as shown in Fig. S4 (SI). The reasoning behind this simplification is twofold. Firstly, the geometry of the problem constrains sound reflected by the bottom followed by the terminus, and sound reflected by the surface between the source and terminus tends to be scattered by the surface waves and bubbles created by the iceberg impact. Secondly, the glacier terminus is rough, resulting in angle-dependent focusing and scattering. Given these complications, which lie beyond the scope of this paper, we have elected to consider only the effect of energy reflected directly from the terminus in comparison with the direct path from source to receiver. As we will show, the greatest effect from this path over the direct path is a 3 dB increase in sound energy and a typical effect is less than 1 dB. These levels are significantly less than the overall effect of the waveguide or inherent scatter in the intensity of sound generated by individual icebergs. Moreover, these estimates probably represent an upper bound because the irregular shape of the terminus will tend to scatter incident sound and decrease its contribution when reflected.

The magnitude of sound reflected from the terminus was calculated using a wavenumber integration technique (see Eq. (4.3.2) in Brekhovskikh and Lysanov, 1982). The terminus surface was assumed to be perfectly flat, and the angle-dependent reflection coefficient was estimated using standard formulas for a fluid-solid interface (e.g., see Eq. (1.61) in Jensen et al., 2011). The compressional and shear wave velocities for the ice were taken to be 3840 m s$^{-1}$ and 1830 m s$^{-1}$, respectively, consistent with those reported by Vogt et al. (2008) for bubble-free ice (a review of the literature failed to reveal sound speed values for bubbly ice below 100 Hz). A range of absorption coefficient values were considered in the analysis; from 0.1 to 1.0 dB $\lambda^{-1}$ for longitudinal waves and from 0.2 to 2.0 dB $\lambda^{-1}$ for shear waves (Rajan et al., 1993; Hobæk and Sagen, 2016). Fig. S4a in Supplementary Information illustrates relationship between the angle of incidence of incoming calving noise and resulting ice reflection loss. Three regions can be identified in this figure: 1. up to 20°, the loss is controlled by the ice-water sound speed ratio and typically reaches a value of approximately 7.5 dB, 2. between 20° and 55°, high attenuation of acoustic energy exceeding 15 dB results mainly from absorption in ice, and finally 3. for larger angles glacier terminus





reflects most of the noise energy back to the water. The analysis demonstrates that the ice reflection loss of calving noise depends greatly on the location of a calving event relative to the glacier-ocean boundary and position of the acoustic buoy.

Further analysis was performed using receiver ranges of 700 m and 1500 m, which correspond to the terminus-receiver ranges for the experiment. The source frequency was set to the middle of the analysis band (50 Hz) and the source position was varied along the terminus at a fixed distance to the ice cliff of 10 m (Fig. S4b in SI). Total energy at the receiver was calculated from the incoherent addition of the direct and terminus-reflected paths and compared with direct path only. The results of this analysis are shown in Fig. S4c. At a range of 1500 m, the maximum contribution of ice-reflected path is always smaller than 1 dB because of the steep angles of incidence. At a closer distance of 700 m, the range of possible angles is extended and a maximum increase in received calving noise of around 3 dB can be expected as a 'worse-case' scenario. Based on these findings, we assumed a typical contribution from ice reflection of 1 dB and corresponding ±1 dB variation around this level.

### 3.5 Impact energy model

The impact energy model requires an estimate of the total sound energy radiated by a calving event, which can be calculated from:

$$E_{\text{ac, imp}} = (E_{\text{ac, obs}} - E_{\text{ac, bckg}})10^{\frac{-TL}{10}}, \tag{5}$$

where $TL$ is a negative number in dB units. The subtraction of $E_{\text{ac, bckg}}$ from the observed impact noise at the hydrophone, $E_{\text{ac, obs}}$, removes background noise energy from the measurement. The factor containing TL transforms the corrected, observed energy into source energy at the impact location using the modeled transmission loss. A transmission loss of $-10$ dB, for example, corresponds to one order of magnitude decrease in received energy. Based on visual inspection of the scatterplot between $E_{\text{imp}}$ and $E_{\text{ac, imp}}$, we used a log$-$log transformation to improve linearity in this relationship. The same type of transformation was revealed by an application of the Box-Cox algorithm, which is often used to normalize regression variables (Box and Cox, 1964). The linear model of conversion between log-transformed energies is given by

$$\ln \hat{E}_{\text{ac, imp}} = a + b \ln E_{\text{imp}}. \tag{6}$$

Having $a = \ln \eta$ and knowing that $b \ln E_{\text{imp}} = \ln E_{\text{imp}}^b$, the power-law relationship has a final form given by:

$$\hat{E}_{\text{ac, imp}} = \eta E_{\text{imp}}^b. \tag{7}$$





Coefficients $a$ and $b$ could be easily derived from a least squares linear regression model using transformed energies as variables. However, both $E_{\mathrm{imp}}$ and $E_{\mathrm{ac,\,imp}}$ have associated uncertainties, which should be considered in a more sophisticated error-in-variables model. Therefore, to address this issue, we used the unified equations for slope, intercept, and associated standard errors proposed by York and others (2004). This approach always gives an answer that is symmetric for both

choices of dependent and independent variables and, after a simple algebraic transformation, Eq. (7) can also be applied to predict the kinetic energy of the ice block from the acoustic emission at its source. Finally, to exclude outliers from the analysis, we identified all points for which uncertainty in acoustic energy calculated with Eq. (5) is not within two standard deviations of the modelled impact noise energy.

## 4 Results and discussion

This section integrates the acoustic and photographic observations of calving events into a power law model that quantifies ice mass loss from the noise energy generated by iceberg impact onto the ocean. The model formation begins with a discussion of the statistics of iceberg volume and drop height estimated from photogrammetry of the time-lapse images, leading to estimates of the block impact kinetic energy (4.1). This is followed with an analysis of the acoustic emission from ice block impacts in terms of its amplitude-frequency characteristics, resulting in an estimate of the total underwater noise

energy generated by a calving event (4.2). The next subsection (4.3) provides a detailed error analysis of these key variables in terms of uncertainty in measurements of the environment, such as bathymetry and thermohaline structure. The power law model relating $E_{\mathrm{imp}}$ and $E_{\mathrm{ac,\,imp}}$ is presented and discussed in (4.4). Finally, an estimate of solid ice discharge determined from the underwater acoustic record using the power law model and integrated over a specific number of calving events is presented (4.5).

### 4.1 The statistics of iceberg volume and drop height

A total of 169 subaerial calving events were captured by time-lapse camera and unambiguously identified with acoustic events (see section 2.2). Individual detachments were unevenly distributed along the whole active part of the Hans Glacier terminus (Fig. 1). The distance to camera, drop height, exposed terminus area, and estimated block volume of the calving

inventory are summarized in Fig. 5.

The distance between camera 1 and the locations of block-water impacts varies from 1700 to 2150 m, with an average of 1880 m (Fig. 5a). The drop height spans 8 to 32 m, with a mean value of $\bar{h} = 18.3$ m (Fig. 5b). The range of exposed terminus is 125 to 5,850 m$^2$ of the ice cliff surface with an average exposed area of 1590 m$^2$ (Fig. 5c). Iceberg volumes

were estimated from $A_{\mathrm{c}}$ using Eq. (2) and vary from $0.2 \times 10^3$ to $53.7 \times 10^3$ m$^3$. The volume distribution is weighted



toward smaller calving events and approximately 90 % of the ice blocks have a volume of less than $20 \times 10^3$ m$^3$ (Fig. 5d). This observation is consistent with previous reports on the power-law distribution of iceberg sizes in Svalbard (Chapuis and Tetzlaff, 2014), Alaska (Neuhaus et al., 2019), Greenland (Sulak et al., 2017), and Antarctica (Tournadre et al., 2016). A least-mean-squares error analysis of the power law distribution of iceberg volumes was made using log-transformed

variables. The best-fit decay exponent of 1.48 (Fig. 5e) found for the present dataset lies between the exponent of 1.69 for Krone Glacier, Svalbard, reported by Chapuis and Tetzlaff (2014) and 0.85 for Perito Moreno Glacier, Patagonia reported by Minowa et al. (2018).

Ice block volume versus drop height is shown in Fig. 5f. The highest iceberg volumes are observed for $h$ within the range of

17 and 26 m, which corresponds well to the mid-heights of the glacier terminus at the locations of calving events. Inspection of Fig 5f shows that ice block volume is correlated with drop height; the Pearson's correlation coefficient is found to be 0.47 and 0.55, respectively for log-transformed and non-transformed variables. This is not altogether surprising because the largest blocks of ice cannot fall from the bottom of the terminus whereas the smaller blocks of ice are not so constrained. The correlation between drop height and iceberg mass is a source of bias in the relationship between ice block volume and

impact kinetic energy and must be accounted for when inverting acoustic recordings of impact noise for ice block volume. This issue is discussed in detail in (4.5).

## 4.2 The generation of underwater sound by iceberg calving

Figure 6 shows a comparison between power spectral density estimates for underwater noise from calving and background

noise recorded by buoys A1 and A2. Spectrograms of the noise generated by a randomly selected calving event are shown in in panels (a) and (b). The computed difference in time of arrival between the two receivers was subtracted from the more distant receiver for better juxtaposition. The two primary sources of sound in the spectrograms are ice melt noise and the underwater noise of calving.

The signal of ice melt, driven by impulsive bubble release (Urick, 1971), is most pronounced between 1 and 3 kHz and corresponds well to the spectral bands reported in previous studies (Deane et al., 2014; Pettit et al., 2015). This signal remains stable during the short observation period. The underwater noise of calving is a by-product of the interaction of the falling iceberg with the ocean. The noise is evident from 2 to 8 s in the recording, at frequencies below 1 kHz. The acoustic intensity varies in both time and frequency. This variability is almost certainly driven by different noise production

mechanisms active at different phases of the calving event (see the high variability in power level between 2 and 4 s, for example). As pointed out by Bartholomaus et al. (2012), low-frequency seismic signals from the impact of ice blocks on the sea surface are generated by 3 major mechanisms: 1. the transfer of momentum from the falling block to seawater, 2. iceberg deceleration due to buoyancy, and 3. the collapse of an underwater air cavity and subsequent emergence of Worthington jets



(e.g. Gekle and Gordillo, 2010). The latter is only possible during total submergence of the ice block, the occurrence of which depends mainly on iceberg dimensions and drop height. Therefore, some calving events may not result in the creation of an air cavity. Moreover, falling icebergs are often fragmented or impact the water at various angles, which certainly modifies all 3 mechanisms of noise production. The influence of calving style on sound emission lies beyond the scope of this work but is likely a significant factor in the variability in sound generation by blocks of similar mass and drop height, as discussed in section 4.4.

The unique patterns in the time and frequency distribution of calving noise potentially contains information about the details of the calving event. For example, Glowacki et al. (2015) hypothesized that subaerial and submarine calving events may generate spectrograms with distinct features, allowing calving mode to be distinguished. Notwithstanding the potential for this kind of analysis, attention here is restricted to a single number, which is the time and frequency integrated energy in the sound field generated by the iceberg impact. Calculation of this number requires selection of the start and stop times of the impact noise and the frequency band over which the noise exceeds background sound levels, which are now considered. The significant increase in noise power accompanying calving allow easy identification of event start and stop times, and these have been selected manually for each event analyzed (see section 2 in SI). The selection of the band of frequencies to consider for the energy calculation is made using background noise levels as a reference. Panels (c) and (d) in Fig. 6 show a 6 s average of noise power spectral density for a calving signal (blue) and background noise recorded just before the event (red). There is a difference between the calving and background noise levels at frequencies up to 700 Hz and 400 Hz for buoys A1 and A2, respectively. The maximum increase in received noise power from calving is approximately 40 dB for both buoys, which corresponds to a factor of 10,000 in acoustic power. The results in Fig 6 show that the appropriate band of frequencies to consider for calving impact noise ends at around 1 kHz. The actual $f_{low}$ was chosen to yield the highest correlation between the impact energy and the received acoustic energy, which was 100 Hz.

The variability of calving noise power across the entire dataset is shown in panels (e) and (f). The normalized power spectral densities of calving events and background noise are plotted as blue and red dots, respectively. A normalization factor is chosen for each calving event and taken to be the highest power level in dB during the event. The same normalization factor is used for both calving and background noise. Calving signatures are clearly distinguishable from the background noise across the entire dataset. However, the calving noise power is noticeably more variable at receiver A2 than A1. This is likely the result of the longer propagation path to A2 and the shallower depth of the hydrophone (22 m at A2 versus 40 m at A1). A shallower waveguide results in more reflections from both sea surface and sea bottom, increasing losses from surface scattering and bottom absorption (e.g. Jensen et al., 2011). The net result of the increased scatter in calving noise observed at location A2 resulted in a decrease in correlation between total impact energy and impact noise, and data from this buoy is not considered further.





### 4.3 Details of error analysis

There are errors in the estimation of both block-water impact energy and impact noise energy. There are two sources of uncertainty for both parameters: measurement error and uncertainty in the state of the changeable environment, which is impossible to characterize completely. Estimates of these uncertainties can be made for the various stages of the analysis
connecting impact noise to ice mass loss and these are discussed below.

### 4.3.1 Uncertainty in block-water impact energy

Assumptions and approximations need to be made when determining the kinetic energy of the falling ice block from time-lapse images. Uncertainties in estimates of the block-water impact energy result mainly from the conversion of the exposed
area at the glacier terminus into ice block volume (see section 3.1). Moreover, additional errors are associated with the details of image analysis, related to the spatial resolution of the time-lapse photography ($\sim 80 - 100$ pixels per terminus height) and imprecise determination of the locations of calving events. The total uncertainty in kinetic energy is difficult to estimate accurately due to several factors, including but not limited to 1. the irregular shapes of the icebergs, 2. poorly-understood site-to-site variability of the scaling factor $C$, and 3. changeable camera orientation relative to the glacier
terminus. However, following Minowa et al. (2018), we assume that the errors in $A_{\text{img}}, d, C$ and $h$ are not larger than $10, 5, 20$ and $5$ %, respectively. Then, since uncertainties in the estimates of ice volumes and drop heights are dependent, the total error bound in the kinetic energy of the impacting ice block is estimated to be approximately $33$ %.

### 4.3.2 Errors in calving-generated acoustic energy

Uncertainties in estimates of the iceberg impact noise result from 3 major sources: 1. spatial and temporal variability of the thermohaline structure in the glacial bay (see Fig. 2 a-d), 2. complicated bathymetry along the propagation path, which depends on the location of calving event (see Fig. 3), and 3. angular and frequency dependence of sound reflection from the underwater part of the glacier terminus (see Fig. S4 in SI). Considering both transmission and reflection losses, the total loss of acoustic energy generated by block-water interaction ranges from 47 to 57 dB (see Fig. S5 in SI), corresponding to a
factor of $10^{-5}$ and $10^{-6}$ in acoustic energy at the source across the entire inventory of calving events. We combined variability in transmission and ice-reflection losses for the entire calving inventory to estimate a representative uncertainty of $33$ % in acoustic energy for each individual calving event at its source.





**4.4 Relationship between the block-water impact and acoustic energy**

Estimating calving ice mass flux from calving noise is based on the idea that these two quantities are correlated. Figure 7 shows a scatterplot of impact noise, $E_{ac,\,imp}$ against impact kinetic energy, $E_{imp}$ for the entire dataset. The broken, black line shows the result of a regression analysis of the power law relationship shown in the figure inset. The acoustic energy

generated by a calving event was calculated from the acoustic pressure time series using Eq. (4) and Eq. (5) with manual selection of integration time (see section 2 in SI) and after low-pass filtering at a cutoff frequency of 100 Hz (see section 4.2). The kinetic energies of the falling ice blocks were derived from Eq. (3) using their masses and drop heights estimated from the camera data (see section 4.1).

The range of energy estimates is large, roughly 2.5 orders of magnitude for both, and there is clearly a strong correlation between the energies across their entire range. The regression coefficient $r = 0.76$ was found between the log-transformed variables for p $< 0.0001$. After removing two outliers and applying an error-in-variables linear regression (see section 3.5 for details), the best functional relationship between acoustic energy and impact energy was found to be a power law relationship given by Eq. (7), where $\eta = 8 \times 10^{-7} \pm 60$ % and $b = 0.92 \pm 3$ %, respectively for the multiplication factor

and exponent of the power law. For completeness, this analysis was repeated including the two identified outliers and the results are shown in Fig. S6 (SI). Glowacki et al. (2015) previously reported $\eta = 5.16 \times 10^{-10}$ and $b = 1$, which gives an impact energy that is 2.5-orders of magnitude higher in comparison to the results presented here (see Fig. S7 in SI). This discrepancy is due to the overly simplified propagation geometry assumed in the earlier study – simple cylindrical spreading loss and no sound reflection from ice terminus – which resulted in an under-estimate of the impact noise energy.

The multiplication factor $\eta$ can be thought of as a conversion efficiency of kinetic energy of a falling iceberg to impact noise energy. The small value of $\eta$ shows that only a tiny fraction of the ice block energy is transformed into underwater sound, which then propagates from the point of impact to the acoustic receiver. A low conversion efficiency is consistent with observations reported for other physical mechanisms of underwater noise generation. For example, only $\sim 10^{-8}$ of the energy

dissipated by a breaking surface wave on the ocean is radiated as sound (Loewen and Melville, 1991). Similarly, the conversion efficiency of the impact energy of a $1 - 5$ mm scale raindrop falling on the sea surface to underwater impact noise is in the range $10^{-9}$ to $10^{-8}$ (see Eq. 4.6 in Guo and Ffowcs Williams, 1991 and Gunn and Kinzer, 1949).

Despite a strong correlation between impact energy and impact noise, there is also a significant scatter in impact noise

energy (roughly a factor of 10) for a given value of kinetic energy. This spread in values can be only partly explained by errors in the energy estimates, which are indicated by blue whiskers in the Fig 7. The scatter is presumably caused by differences in noise generation between individual calving events. The consequence is that estimating the impact energy of an individual calving event from the total noise energy it radiates is accompanied with significant uncertainty. However,





because of the overall good correlation between noise and impact kinetic energy, it is possible to predict the total impact energy summed over a finite number of calving events, provided the inventory is large enough. The uncertainty in individual events tends to average out if enough events are considered, as discussed in section 4.5.

**4.5 Estimation of ice mass loss from the calving noise**

Figure 7 and Eq. (7) show that the relationship between iceberg impact energy and calving noise can be modeled robustly with a power law relationship, providing a means of estimating impact energy from calving noise. Although there is significant variability in doing this on an event by event basis, low-error estimates of cumulative impact energy can be made using Eq. (7) if enough events are added together. Once found, the cumulative impact energy can be converted into an
estimate of iceberg calving flux as follows.

The cumulative modelled ice mass loss from $N$ observed calving events is related to the cumulative impact energy, as inferred from the acoustic signal, by:

$$g \sum_{j=1}^{N} h_j \widehat{M}_j = \sum_{j=1}^{N} \widehat{E}_{\text{imp},j},$$    (8)

where $g$ is the acceleration due to gravity, $h_j$ is the height of the center of mass of the $j^{\text{th}}$ iceberg before separation from the glacier terminus, $\widehat{M}_j$ is the mass of $j^{\text{th}}$ iceberg determined from its underwater impact noise and $\widehat{E}_{\text{imp},j}$ is the kinetic energy of impact of the $j^{\text{th}}$ iceberg. The cumulative ice mass lost through calving would be trivial to compute from Eq. (8) if the mean iceberg drop height was independent of the iceberg mass but this is not the case (see Fig. 5f). Icebergs that extend a
significant fraction of the exposed terminus height have a minimum drop height that is larger than the minimum drop height possible for smaller icebergs. For this (and possibly other) reasons there is a correlation between iceberg drop height and iceberg mass, the consequence of which is that $h_j$ cannot be moved outside the sum on the left-hand side of Eq. (8). The correlation is dealt with by introducing the mass-weighted drop height:

$$\widehat{h} = \sum_{j=1}^{N} h_j \widehat{M}_j \bigg/ \sum_{j=1}^{N} \widehat{M}_j.$$    (9)

It follows immediately from Eq. (8) and Eq. (9) that the cumulative mass sum is given by





$$\sum_{j=1}^{N} \widehat{M}_j = \frac{1}{g\hat{h}} \sum_{j=1}^{N} \hat{E}_{\text{imp},j}, \tag{10}$$

which provides a means of computing the calving flux, since the kinetic energy of iceberg impact can be estimated from its underwater noise using Eq. (7).

We are left with the problem of computing $\hat{h}$, which can be expressed as:

$$\hat{h} \cong \hat{\alpha}\bar{h}, \tag{11}$$

where $\bar{h}$ is the mean drop height and the constant $\hat{\alpha}$ is given by

$$\hat{\alpha} = \lim_{N \to \infty} \left[ \sum_{j=1}^{N} h_j \widehat{M}_j \middle/ \left( \sum_{j=1}^{N} h_j \sum_{j=1}^{N} \widehat{M}_j \right) \right] \approx \lim_{N \to \infty} \left[ \sum_{j=1}^{N} h_j M_j \middle/ \left( \sum_{j=1}^{N} h_j \sum_{j=1}^{N} M_j \right) \right] \tag{12}$$

The final sum on the right-hand side of Eq. (12) is in terms of iceberg mass inferred from the camera observations, providing
a means of computing the mass-weighted drop height on a glacier-by-glacier basis from camera observations. The constant $\hat{\alpha}$ and resulting mass-weighted average drop height are estimated to be 1.13 and 20.7 m for Hans Glacier.

Equation (10) for the cumulative calving mass flux contains significant uncertainty when $N$ is small because of the large scatter in the total underwater sound energy generated by calving events with similar impact energies (see Fig. 7), but the
20 uncertainty reduces as $N$ increases. How large must $N$ be to achieve a desired degree of uncertainty? To answer this question, a Monte Carlo simulation of cumulative ice mass loss was performed using $n$ calving events randomly selected (with replacement) from the entire inventory of calving observations (for which $N = 169$). This selection is repeated $\psi_{\text{max}}$ times ($\psi = \{1, \dots, \psi_{\text{max}}\}$) for each $n = \{1, \dots, n_{\text{max}}\}$, noting that the total number of possible sets of calving events (and associated cumulative kinetic energies and masses) is given by

$$\mathbb{C} = \binom{n + N - 1}{n} = \frac{(n + N - 1)!}{n! (N - 1)!}. \tag{13}$$





From Eq. (10), the cumulative mass sum for a given number of randomly selected calving events $n$ and iteration $\psi$ is:

$$\sum_{i=1}^{n} \widehat{M}_i^{(\psi)} = \frac{1}{g\hat{h}} \sum_{i=1}^{n} \hat{E}_{\mathrm{imp},i}^{(\psi)}, \tag{14}$$

where $\hat{h}$ is calculated from Eq. (11) and Eq. (12) using the $N = 169$ observed calving events. The modelled mass $\widehat{M}_i^{(\psi)}$ in Eq. (14) corresponds to $\widehat{M}_j$ in Eq. (10), where $1 \leq j \leq 169$. The inferred, cumulative ice mass normalized by the cumulative ice mass measured with the camera is then given by

$$\beta_n^{(\psi)} = \sum_{i=1}^{n} \widehat{M}_i^{(\psi)} \Big/ \sum_{i=1}^{n} M_i^{(\psi)}, \tag{15}$$

where $\beta$, for a specified $n$ and averaged over $\psi_{\mathrm{max}}$ iterations, can be expressed as:

$$\bar{\beta}_n = \frac{1}{\psi_{\mathrm{max}}} \sum_{\psi=1}^{\psi_{\mathrm{max}}} \left( \sum_{i=1}^{n} \widehat{M}_i^{(\psi)} \Big/ \sum_{i=1}^{n} M_i^{(\psi)} \right) \tag{16}$$

We set $n_{\mathrm{max}}$ to 1,000 and $\psi_{\mathrm{max}}$ to 10,000 to determine the statistical properties of $\beta$ over a broad range of sample sizes. We note that the probability of randomly obtaining the same set of calving events is vanishingly small for the chosen $\psi_{\mathrm{max}}$ ($\mathbb{C} \gg$ 10,000 for $n \geq 3$, see Eq. (13)).

Figure 8 shows the mean, $\bar{\beta}_n$, and standard deviation, $\beta_{n,std}$, of the statistical distributions of $\beta_n^{(\psi)}$ computed from the Monte Carlo simulation. The correct and unbiased estimate of the mean ice mass flux ratio is $\bar{\beta}_n = 1$, which is indeed the asymptotic value reached for large $n$. As expected, the temporal resolution of the acoustic technique increases with increasing calving activity. The estimated calving flux is within 20 % and 10 % of the expected value when integrating over 40 and 135 ice blocks, respectively (Fig. 8). The number of calving events required for a specified level of ice flux uncertainty translates into an observational timescale that must be met depending on calving rate. For example, at Hans Glacier an uncertainty in ice mass flux of about 20 % is expected when integrating over 2 days of acoustic measurements, corresponding to a calving rate of 20 icebergs per day. The time interval required for a specified level of uncertainty will vary between glaciers and over time. For example, some glaciers calve more than 10 ice blocks hourly (e.g. How et al., 2019) leading to a relatively short time interval requirement.





In addition to a minimum value for sample size for a specified error requirement, there are five other parameters that must be known to compute reliable estimates of ice mass flux from calving noise: the mass-weighted, average iceberg drop height , $\hat{h}$, the conversion coefficient from exposed area to block volume $C$, the conversion efficiency from impact to acoustic

energy, $\eta$, the power law coefficient $b$ and the transmission loss from the glacier terminus to the hydrophone position, $TL$. Errors in these parameters are important because they affect the uncertainty and temporal resolution of the acoustic measurements of calving fluxes (see Figs. 7 and 8).

The problem of how site-specific these parameters are lies beyond the scope of this work and similar studies should be
performed for different tidewater glaciers to obtain quantitative answers. Nevertheless, we briefly discuss here some techniques for measuring or modeling these parameters along with environmental factors driving variability between sites. Noise energy loss is usually calculated using a standard propagation model, such as the Bellhop model used here. Propagation models require sound speed and bathymetry profiles as inputs, making hydrographic and CTD surveys an essential component of the acoustic measurements of calving fluxes. Although the thermohaline structure of a glacial bay is
complex and three-dimensional (e.g. Jackson et al., 2014), patterns of temperature and salinity that are sufficiently characteristic of prevailing conditions in the bay can be identified from limited field measurements and used for propagation model inputs (Glowacki et al., 2016). We anticipate that there is a high uncertainty associated with the conversion coefficient from exposed area to block volume, which likely varies between glaciers characterized by different surface velocity, thermal regime, hydrology, terminus height, etc. This parameter can be determined more accurately for a specific glacier using short-
term lidar measurements or image analysis, e.g. structure-from-motion or stereo photography. These techniques can also provide an estimate of the average drop height. The value of $\hat{h}$ is expected to be close to one-half of the average terminus height (in its active part), because the size of ice blocks breaking off from the top or bottom part of the ice cliff is limited (see Fig. 5f). We hypothesize that the remaining two parameters, the energy conversion efficiency and power law coefficient, are likely stable between glaciers of similar geometry and flow dynamics.

## 5. Concluding remarks

Calving flux from Hans Glacier, Svalbard, has been quantified by analyzing the underwater noise generated by 169 subaerial iceberg-water impacts. The inversion methodology is based on an observed robust ($r = 0.76$), power law relationship between ice block/water impact energy and its resulting acoustic emission below 100 Hz, with an impact-to-noise energy
conversion efficiency of $8 \times 10^{-7}$. The data show that there is significant variability in sound energy production between calving events of similar scale but stable estimates of ice mass flux can be made if enough events are summed (40 events for a 20% standard error at Hans Glacier). The model analysis shows that there are 5 parameters that must be known, as



discussed in section 4.5. It remains to be seen how site specific these parameters are, but transmission loss through the bay and the relationship between exposed area at the glacier terminus and block volume is expected to be variable between glaciers and will likely require site-specific determination. We speculate that the energy conversion efficiency $\eta$ and power law exponent $b$ are likely robust for tidewater glaciers of similar setting.

An important characteristic of any measurement technique is its temporal resolution. While we expect that acoustic determination of ice mass flux will be possible for a broad class of glacier settings, the resolution of calving flux estimates will not be the same for each glacier. The temporal resolution of the acoustic technique for a specified accuracy depends on enough events being observed, so the observation interval is sensitive to calving activity at a particular location. For

example, some tidewater glaciers produce a large number ($> 100$) of small ice blocks daily, while others calve large icebergs ($> 10^8$ m$^3$) not more frequently than every few days (Åström et al., 2014; Chapuis and Tetzlaff, 2014). For the latter, satellite methods are probably the most appropriate when quantifying calving fluxes.

The large inter-event scatter in noise energy generated by ice blocks of similar volume may be reducible. All the information

available in the time and frequency structure of the impact noise (e.g. Fig. 6a) has been reduced to a single number, which is the total acoustic power radiated across a selected frequency band. It is possible that some relevant and variable dynamics of the ice block impact, such as impact angle, block submergence, block integrity, and so on, may leave an identifiable signature in the time-varying frequency structure of the impact noise. If so, then some of the scatter evident in Fig. 7 may be reducible with an improved understanding of the influence of different calving styles and associated source mechanisms on

the received noise spectra. Similar conclusions also arise from seismic measurements (e.g. Bartholomaus et al., 2012). In situ studies of the hydrodynamics of iceberg calving are difficult to imagine in practical terms but scale model laboratory experiments may prove to be a valuable tool in identifying major features of block/water impact dynamics and exploiting their acoustic signatures to reduce uncertainty in the efficiency of noise generation.

**Author contribution**

OG conceived the study and analyzed the data. GBD supported model development. Both authors contributed to manuscript preparation.





**Data availability**

The Bellhop sound propagation model was downloaded from the online Ocean Acoustic Library available at [http://oalib.hlsresearch.com/AcousticsToolbox/]. For image analysis, we used ImageJ software, which can be downloaded free of charge from [https://imagej.nih.gov/ij/download.html]. Bathymetry data were provided by the Institute of

Geophysics, Polish Academy of Sciences who obtained it from the Norwegian Hydrographic Service with permit number 13/G722. Satellite images were downloaded from [https://earthexplorer.usgs.gov/], courtesy of the U.S. Geological Survey, Department of the Interior. All data collected under the monitoring program of the Polish Polar Station Hornsund can be accessed free of charge via the website [https://monitoring-hornsund.igf.edu.pl/index.php/login]. The acoustic data used in this study are available upon request from the corresponding author: oglowacki@ucsd.edu.

**Acknowledgements**

The study was funded by the Ministry of Science and Higher Education of Poland under the 'Mobility Plus' program, grant 1621/MOB/V/2017, and US National Science Foundation, grant OPP-1748265. The field work was supported by the Polish National Science Centre, grant 2013/11/N/ST10/01729, US Office of Naval Research, grant N00014-17-1-2633, and partially within statutory activities 3841/E-41/S/2016 of the Ministry of Science and Higher Education of Poland. We would

like to thank Mateusz Moskalik and Mariusz Czarnul for their significant efforts in maintaining oceanographic and photographic monitoring during the study period. We are also grateful to Aleksandra Stępień and Adam Słucki from the HańczaTech diving team for their underwater work together with Mateusz Moskalik during deployment and recovery of the acoustic buoys, and Kacper Wojtysiak for his work on the development of time-lapse camera systems.

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



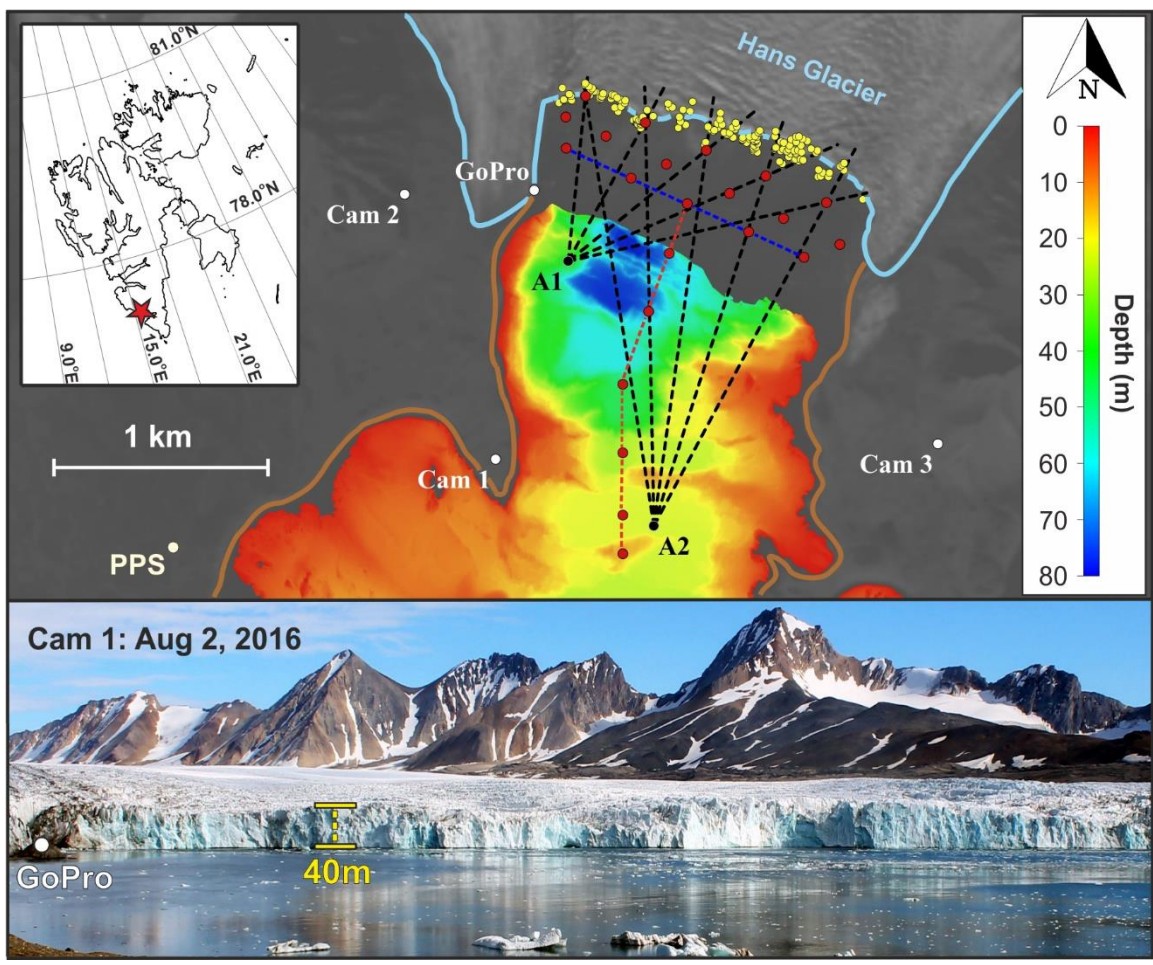

**Figure 1: A map of the study site (upper) and representative cropped time-lapse image taken by camera 1 (bottom).** Top: Locations of time-lapse cameras, acoustic buoys, calving events and CTD casts are marked with white, black, yellow and red dots, respectively.

5    Colored, dashed lines show transects of CTD surveys oriented perpendicular (red) and parallel (blue) to the glacier terminus. Black dashed lines show the spatial arrangement of bathymetry profiles, which we used to model noise transmission losses. Landsat-8 satellite data collected on 27 August 2016, courtesy of the U.S. Geological Survey, Department of the Interior. Bathymetric data provided by the Norwegian Hydrographic Service under the permit no. 13/G722, issued by the Institute of Geophysics, Polish Academy of Sciences.





**Figure 2: (a-d) Sound velocity profiles for CTD surveys oriented perpendicular (red) and parallel (blue) to the glacier terminus, together with (e) the corresponding frequency of calving occurrence.** Locations of the CTD transects taken during the study period are shown in Fig. 1 with the same red and blue colors. Thick, dashed lines mark the dates of the CTD measurements. Blue numbers in the lower panel (e) provide the number of calving events assigned to each set of sound speed profiles.



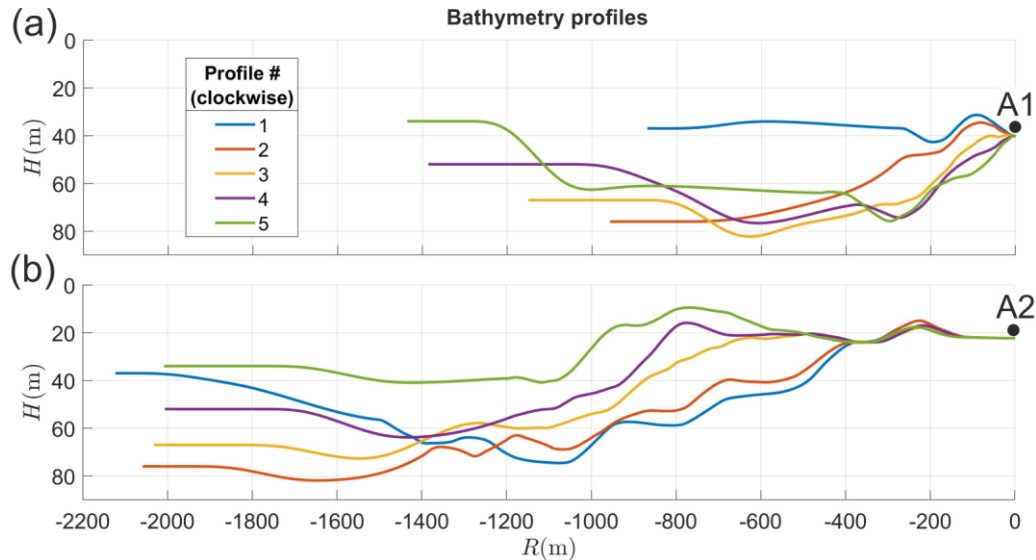

**Figure 3: Bathymetry profiles between the terminus of Hans Glacier and the two acoustic buoys: A1 (a) and A2 (b). The** spatial
arrangement of the transects, which are numbered clockwise, are shown in Fig. 1. The horizontal axis is zeroed at locations of the buoys,
marked with black dots. Bathymetric data provided by the Norwegian Hydrographic Service under the permit no. 13/G722, issued by the
Institute of Geophysics, Polish Academy of Sciences.



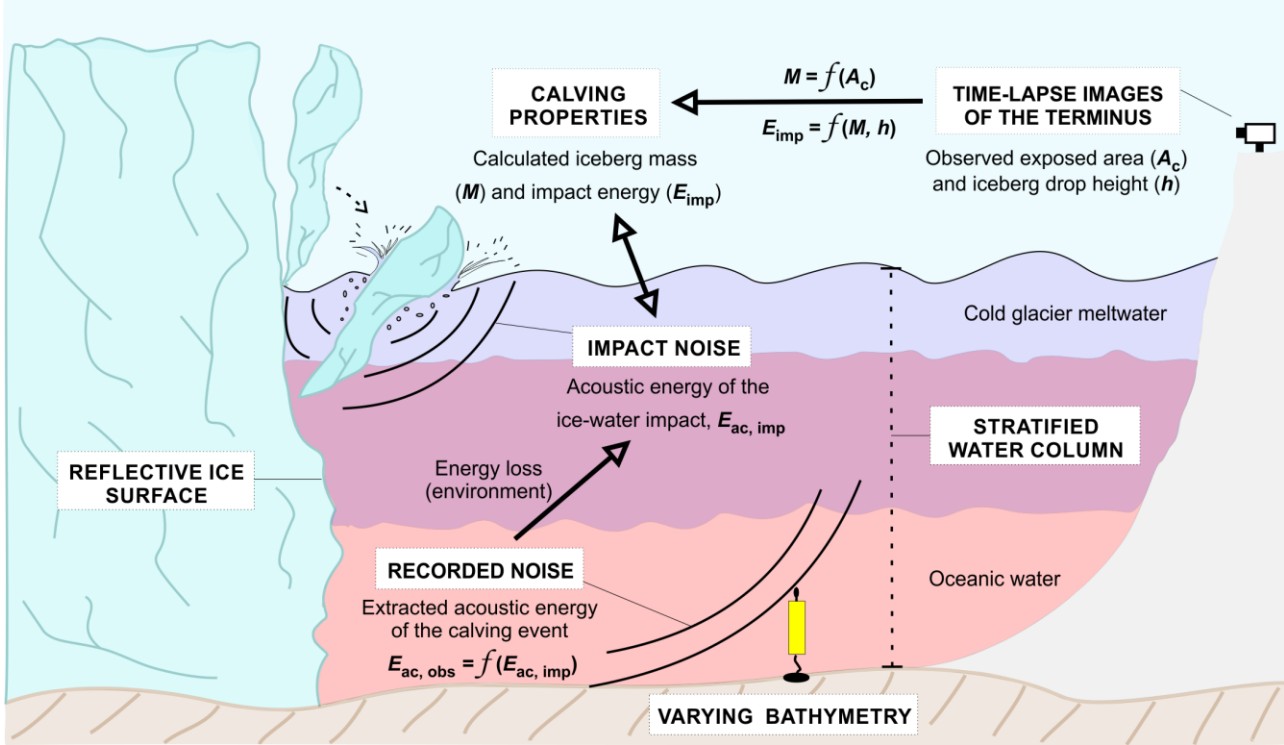

**Figure 4: A scheme illustrating the application of passive underwater acoustics to measure iceberg calving fluxes.** The study consists of 1. time-lapse observation of individual calving events, 2. estimation of ice mass loss and block-water impact energy based on the captured images, 3. recordings of underwater noise at a safe distance from the glacier terminus, and 4. calculation of impact noise energy for given thermohaline conditions, bathymetry along the transmission path and contribution of noise reflected from the ice cliff.



**Figure 5: Histograms of (a) distances between camera 1 and locations of calving events, (b) drop heights, (c) exposed areas of the glacier terminus and (d) estimated iceberg volumes.** (e) Distribution of iceberg volumes divided into 10 bins, presented on log-log scale. The black line shows best-fit power-low (decay exponent $\kappa$) distribution model. (f) Relationship between iceberg drop height and volume. The Pearson correlation coefficient is 0.47 and 0.55 for log-transformed and non-transformed variables, respectively.



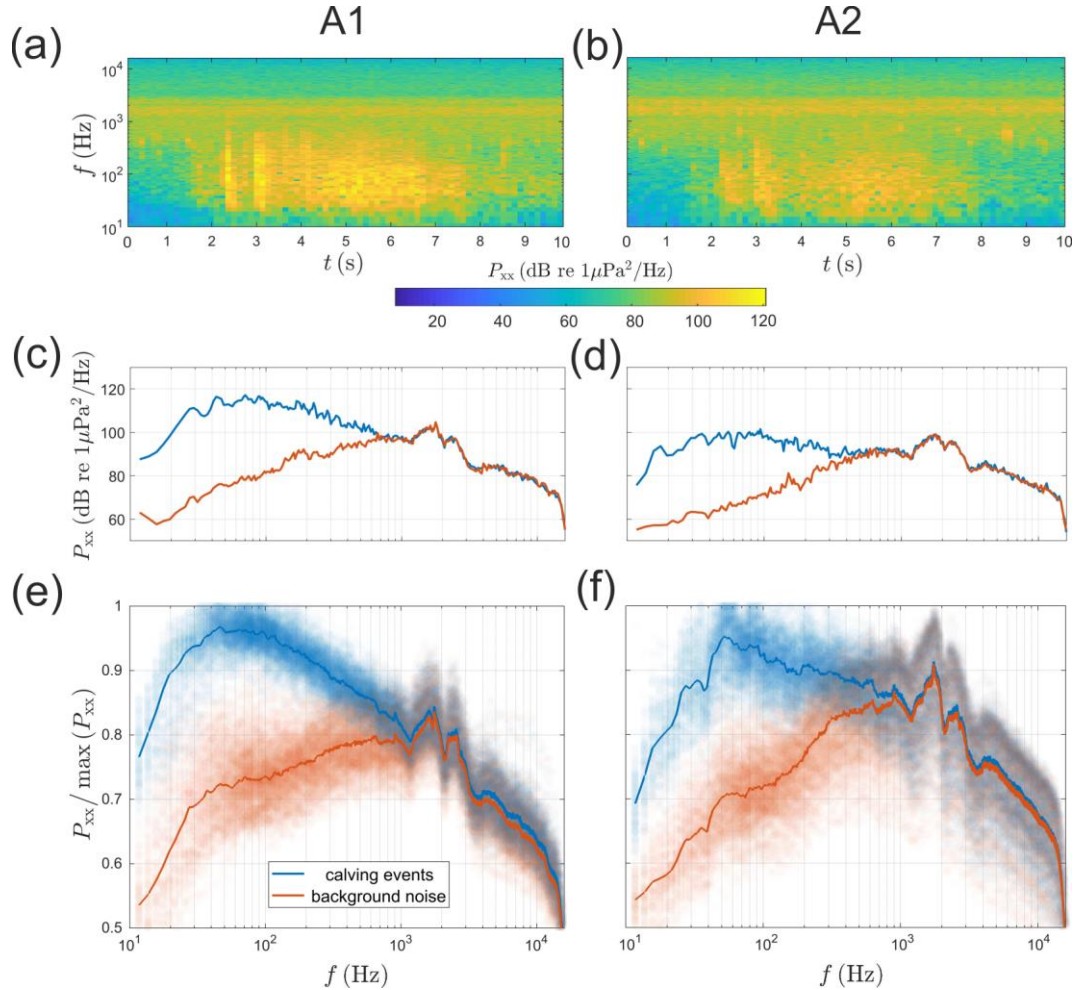

**Figure 6: (a – b) Spectrograms of the acoustic signal generated by the calving event recorded at A1 (left) and A2 (right), (c – d) corresponding time-averaged spectra of background (red) and calving (blue) noise, and (e – f) normalized power spectral densities for the entire calving inventory.** A difference of 10, 20 and 40 dB in $P_{xx}$ corresponds, respectively, to a factor of 10, 100 and 10,000 in acoustic energy. Noise spectra were normalized using maximum values of the calving signal for each event. Solid lines in (e –f) show median normalized spectra.

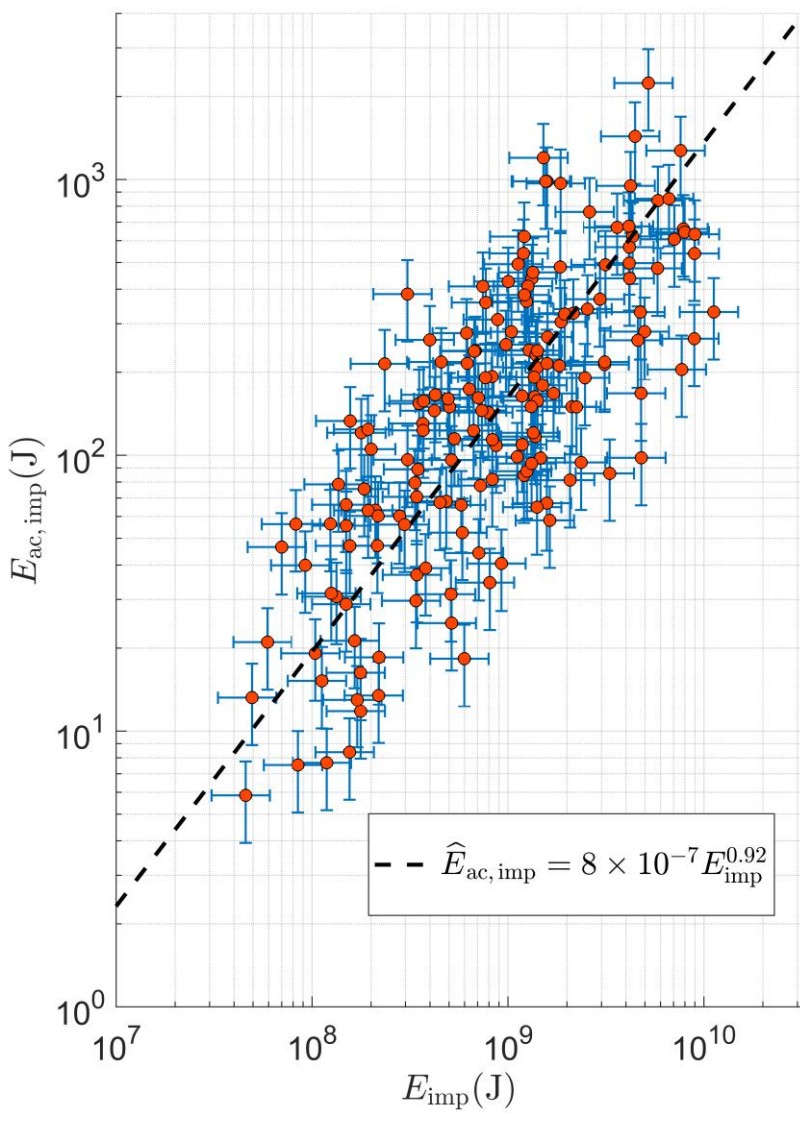

**Figure 7: Relationship between the block-water impact energy and underwater acoustic emission below 100 Hz.** Uncertainties are marked with blue whiskers and were estimated to be 33% for both variables. The remaining scatter in impact energy is most likely caused by different calving styles and an associated variability in source mechanisms. The results with inclusion of outliers are shown in Fig. S6 in Supplementary Information (see text for details).





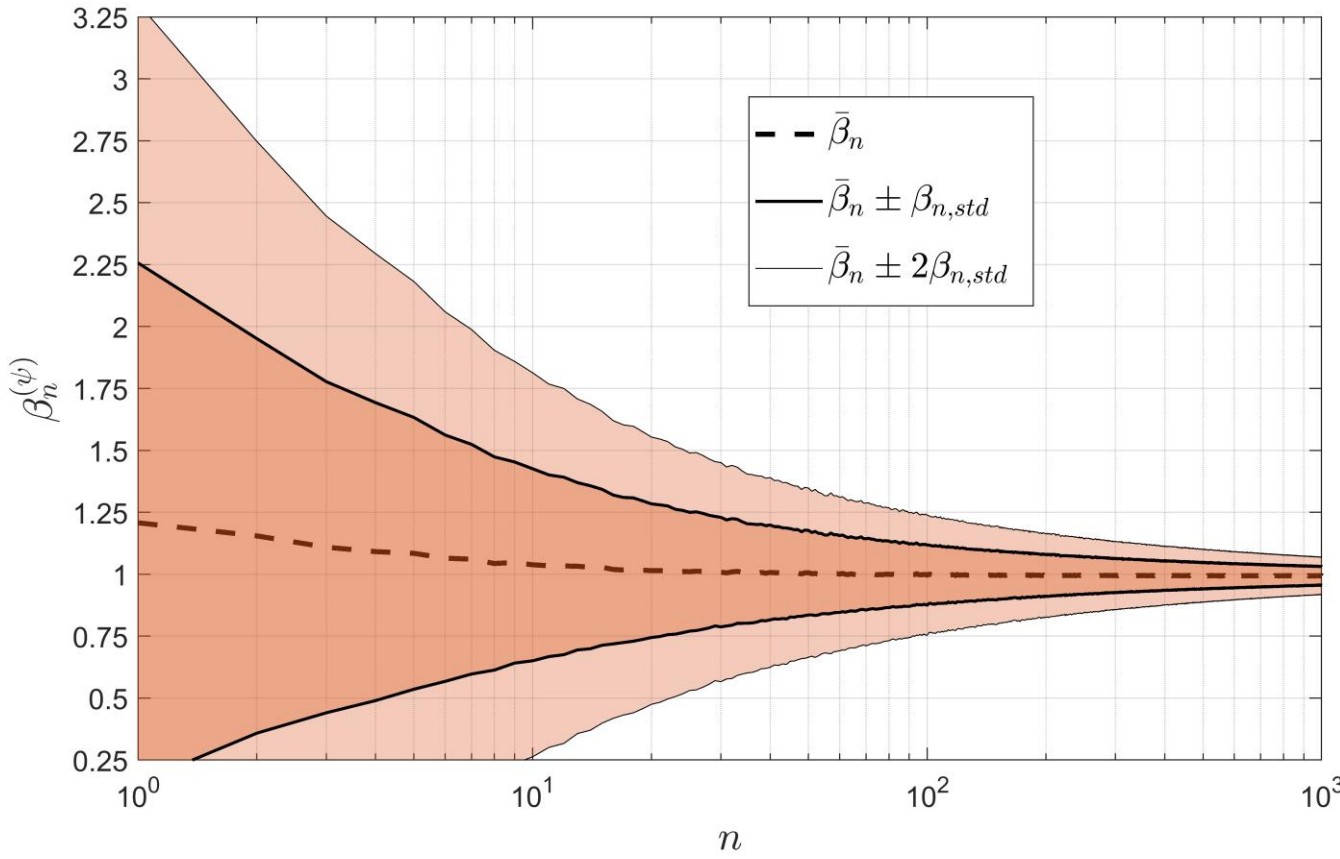

5    **Figure 8: The ratio between modelled and observed cumulative ice mass loss computed using the Monte Carlo method with $n$ events randomly selected (with repetition) from the entire calving inventory.** The selection procedure was repeated $\psi_{max} = 10,000$ times for each $n$. See text for details of the modelled ice mass loss calculation. The thick and thin solid lines respectively denote the 1 and 2 standard deviation boundaries of the distributions.