# Peer review of "Quantifying iceberg calving fluxes with underwater noise"

_The Cryosphere, 2019_

## Referee Comment (RC1) · Andreas Köhler (Referee) · 12 Dec 2019

General comments:

Glacier calving can be observed indirectly through records of underwater-acoustic, infra-sound, seismic, and ocean waves. The key advantage of monitoring these signals compared to (optical) sensing methods is the high temporal resolution and independence from visibility. However, only a few studies so far have attempted to quantify iceberg mass or volume using the observed calving signals. This can be done for example by utilizing empirical models which relate signal properties and calving event sizes. Therefore, this study is a welcome and timely contribution to the development of new methods suitable for monitoring the calving flux of tidewater glaciers. I appreciate

in particular the incorporation of physical models to correct for the effect of wave propagation on the recorded signals, and the detailed discussion of uncertainties. There are a few issues which need clarification and maybe some additional work, but I do not have any major objections that would prevent publication of this well-written article.

Specific comments:

(1) I miss the comparison of the relation between measured acoustic energy and impact energy with and without correcting for the propagation effect. If bathymetry information and sound speed profiles would not be available for Hansbreen, how much worse would the correlation coefficient of log energies be for the same data set? Does the correction also increase correlation for location A2 which you did not consider further? Also, what would be the correlation coefficient if the signal duration instead of total energy of the signal is compared? Showing how taking into account propagation effects on amplitudes actually reduces the scatter in the log energy relation would better emphasize the importance of incorporating local bathymetrical and sound speed profile data. However, since such information might not always be available, it would be interesting to see how much more uncertain or biased the calving flux estimate would be.

(2) In section 4.5 the authors describe how the cumulative ice loss and its uncertainty can be estimated. I miss the actual application of this approach to all hydro-acoustic calving signals in the entire record. This could then potentially be compared to independent measurements of the calving flux at Hansbreen. As far as I understand, the signals of the 169 matched events are only used for model calibration. How many calving signals in total occurred in the entire record, including those without camera observations and unsuccessful matching? If this information is not available, is there any particular reason why the entire data set has not been screened for all calving signals?

(3) How well suitable is the method for long-term continuous monitoring? I encourage

to discuss briefly aspects such as (real-time) data retrieval, power supply, instrument clock drift and regularly updating sound speed profiles. Could detecting calving signals, including picking start and end time of events, be automated? Are there signals of other origins which could be mistaken for calving? Do you see any benefit from using in addition data from the permanent seismometer at the Hornsund station?

Technical corrections:

I personally prefer writing "Hansbreen" over "Hans Glacier" since this is an official place name in Svalbard.

Page 1 line 15: "temporal fluctuations" or better "time dependency of the thermohaline structure"

Page 1 Line 17: Writing "the corresponding measured acoustic energy corrected for these three factors" or something similar would be more precise.

Page 1 Line 19: remove "as we demonstrate"

Page 1 Line 21: "50% uncertainty is . . ."

Page 2, Introduction: It is worth mentioning here that several studies found submarine melting to be the dominant contributor to frontal ablation.

Page 6 line 30: Please explain "calibration of camera geometry" Do you refer to georeferencing the images? How precise is the pixel-area conversion (Equation 1)? Do you take into account the orientation of the calving front with respect to the camera?

Page 7 line 3: "The newly exposed area" or "Newly exposed areas" (same in following sentence)

Page 8 line 6: This sentence describes that PSDs are calculated, but later you do not refer to Pxx again in the text. From this paragraph it is also not clear what PSDs are used for, because in the next sentence you already describe a different processing step, i.e., how the energy is computed in the time domain. I suggest to refer to Figure

6 and/or to write that PSDs are computed to investigate the spectra of events. If Pxx is just used as label in the figure, you do not need to introduce it as a variable in the text.

Page 8 line 26: Please clarify what is meant by "32, 46, 61 and 30 detachments" Is it the day in the measurement period which you group into 4 time segments? What did motivate this division? According to Figure 2, the CTDs are not centered in the middle of each time segment. How fast do you expect the sound profile to vary over time?

Equation 5: For clarity, you could write TL in terms of the modelled factors described in the previous section (terminus reflections, wave propagation).

Page 12 line 3-6: I was not able to follow these sentences. Could you provide more details about the "error-in-variable model"? I do not understand what you mean with "Eq (7) can also be applied"?

Page 12 line 17: Do you actually present an estimate for the solid ice discharge for Hansbreen (see comment 2)? You present the method, but as far as I understand you did not apply it to data. What do you mean with "specific number"? Here you should make clear if you refer to the cumulative total number of calving signals in the considered time period or the different number of calibration events that you use for estimating the calving flux uncertainty (in percent).

Page 13 line 6: Krone Glacier -> Kronebreen

Page 14 line 13: Remove "which are now considered"

Page 14 line 21: Rephrase sentence. Instead of "The actual f_low" write "The lowest frequency was . . . "

Page 14 line 31: Explain "net result". Could you show this decrease in correlation in a figure? Isn't your model supposed to correct for the longer propagation path? See also comment 1.

Page 15 line 14: What do you mean with "changeable camera orientation"?

Page 18 Equation 11 & 12: Can you explain in a bit more detail where these equations come from and why there are valid?

Page 20 line 27: "Calving flux has been quantified": As written earlier, this can be misleading. Impact energies are quantified for individual events and a methodology for quantifying the calving flux is suggested. However, no actual calving flux estimates are computed for Hansbreen because this would require a complete hydro-acoustic calving record (see comment 2). Please clarify.

Page 20 line 28: I am not sure if "inversion" is the correct word here.

Page 20 line 29: Do you mean above 100 Hz? As I understood it, this is the lower frequency limit.

Page 26 line 19: This paper has just been published in The Cryosphere.

Figure 5: How representative is the calving size distribution in (e) with respect to the actual size distribution at Hansbreen? Since this is based on the 156 events unambiguously matching with acoustic signals, could some size range be under- or over-represented due to size-dependent matching ability?

---

## Referee Comment (RC2) · Anonymous Referee #2 · 26 Dec 2019

Underwater acoustics is a thrust area in the oceanography field. Arctic ambient noise analysis is much needed for studying global warming under noise mechanisms techniques. In this paper, authors trying to quantify the iceberg calving fluxes with noise. The manuscript is written very vaguely. A lot of irrelevant information's and very basics are repeating. Particularly introduction part needs reduction. It would be good if the author mentioned about anthropogenic effects during the measurement period. How the shipping noise are filtered and what method utilized. If the iceberg calving events and shipping noise occurred parallelly, special needs to be taken for analysis. I never see any branch of science under "ambient noise oceanography". It should either oceanography or physical oceanography, change the term accordingly. Perhaps the author can change to "Ambient noise analyses". It would be better if the author provide

some photographs showing iceberg calvings. Please check for repeated sentences (for e.g., bathymetry profile for two acoustic buoys). Mooring information is needed (eg. Taut mooring or some other), perhaps a mooring diagram is good to present. How the author quantifies the noise datasets pertaining to only iceberg calving events. The introduction part is too large. PP No. 3, Line 20-30, I don't think it is necessary for mentioning very basics. PP No. 2, Line 20-25 are repeating at PP No. 3, Line 25-30. I suggest the author reduce the introduction part significantly. PP. No. 6 Calibration methods of camera geometry? PP. No. 7 Line 10, references required for the equation (2). PP. No. 11 Line 15, references required and please elaborate it rather than simply writing as "calculated from". Figure 6, labels are very small. What is the source for the frequency band observed at 1kHz (Figure a both A1 and A2)? PP. No. 13. Line 20. Iceberg calving events date and time stamp of occurrence are required. PP. No. 14. Lin 10. The part needs to be moved to the introduction but I don' think it is necessary. PP. No. 21. Mentioning references in the conclusion part is not necessary.

---

## Author Comment (AC1) · 25 Jan 2020

We are grateful to Andreas Köhler for the insightful and very helpful comments on our manuscript. Below are point-by-point replies for all questions and suggestions.

5

(1) I miss the comparison of the relation between measured acoustic energy and impact energy with and without correcting for the propagation effect. If bathymetry information and sound speed profiles would not be available for Hansbreen, how much worse would the correlation coefficient of log energies be for the same data set? Does the correction also increase correlation for location A2 which you did not

- 10 consider further? Also, what would be the correlation coefficient if the signal duration instead of total energy of the signal is compared? Showing how taking into account propagation effects on amplitudes actually reduces the scatter in the log energy relation would better emphasize the importance of incorporating local bathymetrical and sound speed profile data. However, since such information might not always be available, it would be interesting to see how much more uncertain or biased the calving
- 15 flux estimate would be.

These are very interesting questions and the answers are summarized in Table 1 below. In short, the best correlation between iceberg impact energy inferred from the camera data and the acoustic signal comes from the signal that is compensated for transmission loss. The next best correlate is uncompensated signal followed by signal duration (discussed further below). We also note that an additional purpose of the transmission loss calculations is to determine estimates of the power-law coefficients *b* and  $\eta$ , which we suppose to be site-independent. In other words, the model could be tested for other glaciers, if enough information is available on bathymetry and thermohaline structure of the water masses.

25

Table 1 presents correlation coefficients between log-transformed variables. The correlation between impact energy and noise energy for location A1 increases from 0.71 to 0.76 with the inclusion of transmission losses. However, it does not improve correlation for location A2. If the signal duration,

1

 $\Delta t$ , instead of total energy of the signal is used, the correlation drops to 0.61 (see also Fig. 1). The duration of the received calving signal should not depend greatly on the propagation conditions in the bay, if the signal-to noise ratio is high enough. Therefore, it might be possible to use  $\Delta t$  instead of the corrected noise energy to estimate calving fluxes (see best-fit line in Fig. 1R), as an alternative when no

5 information is given on bathymetry and sound speed profiles. Thank you for the suggestion. Figure R1 and Table 1 will be added to the supplement of the revised manuscript. We will also refer to these additional results in the main text.

10 **Table 1** Correlation coefficients between log-transformed variables. The duration of the calving signal is the same for both locations – A1 and A2.

|                                       | E imp | Δt   | E ac, obs
( A1 ) | E ac, imp
( A1 ) | E ac, obs
( A2 ) |
|---------------------------------------|------------------|------|--------------------------------|--------------------------------|--------------------------------|
| E ac, imp
( A2 ) | 0.30             | 0.15 | 0.64                           | 0.53                           | 1                              |
| E ac, obs
( A2 )        | 0.31             | 0.15 | 0.64                           | 0.52                           |                                |
| E ac, imp
( A1 )        | 0.76             | 0.39 | 0.97                           |                                |                                |
| E ac, obs
( A1 )        | 0.71             | 0.34 |                                |                                |                                |
| Δt                                    | 0.61             |      | -                              |                                |                                |

Figure R1: Relationship between the block-water impact energy and the duration of the calving signal.

5

(2) In section 4.5 the authors describe how the cumulative ice loss and its uncertainty can be estimated. I miss the actual application of this approach to all hydro-acoustic calving signals in the entire record. This could then potentially be compared to independent measurements of the calving flux at Hansbreen. As
far as I understand, the signals of the 169 matched events are only used for model calibration. How many

calving signals in total occurred in the entire record, including those without camera observations and unsuccessful matching? If this information is not available, is there any particular reason why the entire data set has not been screened for all calving signals?

- 5 We agree that this methods study does not show a full practical application of the proposed sensing technique. We are in the process of comparing independent measurements of calving flux from Hansbreen with ice mass losses estimated using a continuous acoustic record and both satellite and camera observations of the glacier using a more extensive data set than the calibration observations reported here. We will have enough room in this additional study to cover in detail some important aspects raised in
- 10 comment #3, like automatic detection of calving events from the continuous acoustic recording, for example.

(3) How well suitable is the method for long-term continuous monitoring? I encourage to discuss briefly aspects such as (real-time) data retrieval, power supply, instrument clock drift and regularly updating sound speed profiles. Could detecting calving signals, including picking start and end time of events, be automated? Are there signals of other origins which could be mistaken for calving? Do you see any benefit from using in addition data from the permanent seismometer at the Hornsund station?

15

We have collected several year-long datasets (unpublished to date, see point 2 above) and can modify the 20 manuscript to provide readers with some practical details about how to collect such data, including issues such as power requirements, instrument clock drift and requirements and requirements for taking sound speed profiles. This technique could be made real-time with a cabled or wireless link to shore but the technology required for this capability is untested by us. Calving events are easily and clearly distinguishable from the noise of ice melting in spectrograms of the acoustic record at frequencies below

25 1 kHz We are investigating the automatic detection of calving events (see comment above) but are not ready to discuss this work here.

There are other sound sources that could be mistaken with calving events. For example, calving or the disintegration of bigger icebergs could be mistaken with glacier calving events. Moreover, the calving style/type can be a problem. For example, distinguishing between subaerial and submarine calving events, at least with present state of knowledge. We will be in a better position to tackle these issues when

- 5 considering the analysis of a long-term record. We do see benefits from using both seismic and acoustic measurements in one location. For example, applying both methods could potentially reveal new important information for the formulation of new (or improved) calving laws and seismic signals may help distinguish between the various kinds of events discussed above. It is quite likely that some mechanisms involved in the calving process are detectable/researchable by only one of these techniques.
- 10 We believe that these two methods, cryoseismology and acoustical oceanography, could complement each other to study the tidewater glacier dynamics.

I personally prefer writing "Hansbreen" over "Hans Glacier" since this is an official place name in 15 Svalbard.

Agreed, we will change to the official place name Hansbreen.

20 Page 1 line 15: "temporal fluctuations" or better "time dependency of the thermohaline structure"

We will change to "time dependency of the thermohaline structure", as suggested.

**25**

Page 1 Line 17: Writing "the corresponding measured acoustic energy corrected for these three factors" or something similar would be more precise.

We will make the suggested change.

Page 1 Line 19: remove "as we demonstrate"

**5**

Done.

Page 1 Line 21: "50% uncertainty is . . ."

**10**

Done.

Page 2, Introduction: It is worth mentioning here that several studies found submarine melting to be the dominant contributor to frontal ablation.

Point taken, we will add a new sentence to the introduction: On the other hand, several studies found that increased submarine melting is a main factor responsible for the observed rapid retreat of tidewater glaciers (e.g. Straneo and Heimbach, 2013; Luckman et al., 2015; Holmes et al., 2019).

20

Page 6 line 30: Please explain "calibration of camera geometry" Do you refer to georeferencing the images? How precise is the pixel-area conversion (Equation 1)? Do you take into account the orientation of the calving front with respect to the camera?

25

Our wording here was imprecise. We do not refer to georeferencing the images. We will change this part of the sentence accordingly: The irregular shape of the ice cliff provided registration features, which were identified in both Landsat-8 satellite images (with resolution of 15 m) and the camera images, enabling a precise localization of calving events.

It was impossible to calculate the exact angle between the camera and individual segments of the glacier

- 5 terminus. However, we took this into account by assuming uncertainty in  $A_{img}$  of 10%. The total uncertainty in  $A_c$  (Eq. 1) is then 14%. We will add new text to explain that: The camera was oriented roughly perpendicular with respect to the calving front (Fig. 1), but precise calculation of the exact angle was impossible due to the limited resolution of the satellite images and large variability of the terminus shape over the study period. Nevertheless, this uncertainty was included in the error analysis (see section
- 10 4.3 for details). In subsection 4.3.1 we will provide 14% as an uncertainty in  $A_c$ .

Page 7 line 3: "The newly exposed area" or "Newly exposed areas" (same in following sentence)

15

Done.

Page 8 line 6: This sentence describes that PSDs are calculated, but later you do not refer to Pxx again in the text. From this paragraph it is also not clear what PSDs are used for, because in the next sentence you already describe a different processing step, i.e., how the energy is computed in the time domain. I suggest to refer to Figure 6 and/or to write that PSDs are computed to investigate the spectra of events. If Pxx is just used as label in the figure, you do not need to introduce it as a variable in the text.

25 We agree that this was a bit confusing. We will add the suggested text: (...) to investigate the noise spectra (see Fig. 6). We will remove Pxx from text, leaving it only as a label in the Fig. 6, as suggested.

Page 8 line 26: Please clarify what is meant by "32, 46, 61 and 30 detachments" Is it the day in the measurement period which you group into 4 time segments? What did motivate this division? According to Figure 2, the CTDs are not centered in the middle of each time segment. How fast do you expect the sound profile to vary over time?

5

There were 4 days of CTD measurements during the study period, made with non-constant time intervals: August 1, 12, 30 and September 9, 2016. For each measurement day, we calculated median sound speed profile from the whole set of profiles. Each calving event was then paired with the closest median profile to provide our best estimate for the propagation conditions in the water column. We will clarify this by

- 10 saying: Firstly, a median sound speed profile was calculated from each set of profiles measured at the same day. Then, a closest median profile was assigned to each calving event according to the time of its occurrence. As a result, four consecutive median sound speed profiles were assigned to 32, 46, 61 and 30 calving events, respectively (see Fig. 2).
- 15 The upper panel of Fig. 2 provides some insight into how the sound speed profiles in the bay change in space and time. Large variability is expected, due to the estuarine circulation, eddies, melting of the terminus and drifting icebergs, calving-driven movement of the water masses, freshwater discharge from the glacier, etc. It is impossible/impractical to include all these factors in the propagation model. However, limited CTD measurements reveal the evolution from the relatively depth-independent sound speed to the
- 20 upward refracting profile, with the freshwater lens near the surface (Fig. 2).

Equation 5: For clarity, you could write TL in terms of the modelled factors described in the previous section (terminus reflections, wave propagation).

25

This is a very good point, instead of the TL we will introduce  $TL_{tot} = TL_{prop} + TL_{refl}$ .

Page 12 line 3-6: I was not able to follow these sentences. Could you provide more details about the "error-in-variable model"? I do not understand what you mean with "Eq (7) can also be applied"?

5 We will rephrase to clarify: However, both  $E_{imp}$  and  $E_{ac, imp}$  have associated uncertainties, which should be accounted for in the analysis. Therefore, to address this issue, we used the unified equations for slope, intercept, and associated standard errors proposed in a model by York and others (2004). This model belongs to the family of errors-in-variables regression models, which include all uncertainties and always give an answer that is symmetric for both choices of dependent and independent variables.

10

The phrase "Eq (7) can also be applied" will be removed.

Page 12 line 17: Do you actually present an estimate for the solid ice discharge for Hansbreen (see

- 15 comment 2)? You present the method, but as far as I understand you did not apply it to data. What do you mean with "specific number"? Here you should make clear if you refer to the cumulative total number of calving signals in the considered time period or the different number of calibration events that you use for estimating the calving flux uncertainty (in percent).
- 20 As pointed out before (comment 2), this is a methods paper and no long-term estimates of calving flux are presented. Instead of "Finally, an estimate of solid ice discharge determined from the underwater acoustic record using the power law model and integrated over a specific number of calving events is presented (4.5)." we will write: Finally, based on this relationship, a new methodology is suggested for quantifying the calving flux from the underwater noise of iceberg-water impact (4.5).

25

Page 13 line 6: Krone Glacier -> Kronebreen

**Done.**

Page 14 line 13: Remove "which are now considered"

**5**

**Done.**

Page 14 line 21: Rephrase sentence. Instead of "The actual f\_low" write "The lowest 10 frequency was . . . "

**and related:**

Page 20 line 29: Do you mean above 100 Hz? As I understood it, this is the lower frequency limit.

We mean below 100 Hz, because the low-pass filter was applied. We will change the variable name from  $f_{low}$  to  $f_c$  to clarify this. The subscript 'low' was a bit misleading.

20 Page 14 line 21: We will rewrite to: However, an upper frequency limit of 100 Hz was applied in further analysis to yield the highest correlation between the impact energy and the received acoustic energy.

Page 14 line 31: Explain "net result". Could you show this decrease in correlation in a figure? Isn't your model supposed to correct for the longer propagation path? See also comment 1.

25

15

We will remove the unnecessary "net result" phrase.

Figure R2 (below) show scatterplots between the kinetic energy and impact noise energy computed for both locations: A1 and A2. The correlation decreases and the scatter increases when we use acoustic data from location A2 instead of A1 (see also Table 1). Moreover, the regression line of the power-law fit is different (Fig. R2). There are specific reasons behind this discrepancy.

5

Firstly, the signal-to-noise ratio for acoustic recordings taken at location A2 is lower. The receiver at A2 was more exposed to the contribution from the sound sources located outside the glacial bay of Hansbreen (see Fig. 1 in the main manuscript). Moreover, the mooring was located in very shallow water (~22 m), which means that very low-frequency sound (

---

## Author Comment (AC2) · 25 Jan 2020

**We thank the Anonymous Referee #2 for providing comments on our manuscript. Below are point-by-point replies to these concerns.**

It would be good if the author mentioned about anthropogenic effects during the measurement period. How the shipping noise are filtered and what method utilized. If the iceberg calving events and shipping noise occurred parallelly, special needs to be taken for analysis.

10 An observer present in the field throughout the data collection phase provides verification of the fact that no anthropogenic sound sources were active during the occurrence of calving events analyzed in this study. We did not use any detection algorithms to automatically detect calving events from the continuous acoustic dataset. This is on hold for the future study, where the calving flux from Hansbreen will be estimated, based on the methodology presented here.

15

I never see any branch of science under "ambient noise oceanography". It should either oceanography or physical oceanography, change the term accordingly. Perhaps the author can change to "Ambient noise analyses".

20

We agree that "ambient noise oceanography" could sound unfamiliar for the readers. We will change "ambient noise oceanography" to "acoustical oceanography", which was introduced in the late '70 by Clay and Medwin (1977).

25

It would be better if the author provide some photographs showing iceberg calvings.

In the revised manuscript we will refer to our previous paper, where the videos of different calving events are presented in the supplement (Glowacki et al., 2015).

5    Please check for repeated sentences (for e.g., bathymetry profile for two acoustic buoys).

We will check for repetitions and remove words if needed.

10   Mooring information is needed (eg. Taut mooring or some other), perhaps a mooring diagram is good to present.

We used simple, light moorings, which were recovered by divers. A single mooring system consisted of an anchor, short line and acoustic buoy with positive buoyancy. A short description of the mooring
15   configuration will be added to the methods section of the revised manuscript.

How the author quantifies the noise datasets pertaining to only iceberg calving events.

20   We synchronized the acoustic recordings with camera images. Calving events were clearly and easily distinguishable from other noise sources because ice melt noise dominates the signal at frequencies between 800 Hz and 5 kHz, which is above of range of interest, and there were no ships, marine mammals or other sources of interference. See also answer for the comment #1.

25

The introduction part is too large. and I suggest the author reduce the introduction part significantly.

Our intension is to reach a broad scientific community and using of oceanographic tools to study glacier dynamics is an interdisciplinary endeavor. We want to ensure that glaciologists have a good introduction to acoustical oceanography, which they may not be familiar with. On the other hand, most acousticians are probably not familiar with the vocabulary and methods used in glaciology. The introduction is divided into sections to allow readers to chose whether to go through the specific subsections or not, which we believe will depend on their background knowledge and scope of interest. However, we understand the reviewer's concern on the paper length and we will carefully screen the introduction again for any unnecessary information.

PP No. 2, Line 20-25 are repeating at PP No. 3, Line 25-30.

We are not following this comment. The first segment (P2, L20-25) concerns difficulties in measuring calving fluxes, while the second one (P3, L25-30) focuses on some advantages offered by acoustical oceanography (low cost of the receivers; easy deployment; gathering continues data, even at harsh conditions). We believe that both topics should be covered in the introduction section.

PP. No. 6 Calibration methods of camera geometry?

Further information about camera geometry and image calibration have been provided in response to the similar comment from Reviewer 1.

PP. No. 7 Line 10, references required for the equation (2).

There are two references already given for this equation (Åström et al., 2014; Pętlicki and Kinnard, 2016). See Line 13, page 7. The equation is also supported by the 3rd reference to Dowdeswell and Forsberg (1992). See Line 9, page 7.

PP. No. 11 Line 15, references required and please elaborate it rather than simply writing as "calculated from".

The components of the equation are discussed in detail in the following text, see P11 L17-20. However, 10 we agree with the Reviewer that adding a reference to a standard book on acoustical oceanography (Clay and Medwin, 1977) will be appreciated by the future readers. We will add it in the revised version of the manuscript.

15 Figure 6, labels are very small.

Agreed. An improved version of this figure is shown below.

[Figure]

5    What is the source for the frequency band observed at 1kHz (Figure a both A1 and A2)?

The peak observed in the frequency band (1-3) kHz is associated with the melt noise. See L25-27, P13 for more details and references.

PP. No. 13. Line 20. Iceberg calving events date and time stamp of occurrence are required.

Agreed. We will add date and time stamp for the calving event for which the spectrogram and spectrum is shown on panels a-d, Fig. 6.

PP. No. 14. Line 10. The part needs to be moved to the introduction but I don' think it is necessary.

Agreed. We will remove this sentence.

PP. No. 21. Mentioning references in the conclusion part is not necessary.

We believe that these references will help the readers to follow the conclusions and would like to retain them.

References

Åström, J. A., Vallot, D., Schäfer, M., Welty, E. Z., O'Neel, S., Bartholomaus, T., Liu, Y., Riikilä, T., Zwinger, T., Timonen, J., and Moore, J. C.: Termini of calving glaciers as self-organized critical systems, Nat. Geosci., 7, 874–878, https://doi.org/10.1038/ngeo2290, 2014.

Clay, C. S., and Medwin, H.: Acoustical oceanography: principles and applications, Wiley, New York, USA, 1977.

Dowdeswell, J. A., and Forsberg, C. F. The size and frequency of icebergs and bergy bits derived from tidewater glaciers in Kongsfjorden, northwest Spitsbergen, Polar Res., 11(2), 81–91, https://doi.org/10.3402/polar.v11i2.6719, 1992.

5  Glowacki, O., Deane, G. B., Moskalik, M., Blondel, Ph., Tegowski, J., and Blaszczyk, M. (2015), Underwater acoustic signatures of glacier calving, Geophys. Res. Lett., 42, 804–812, https://doi.org/10.1002/2014GL062859, 2015.

Pętlicki, M., and Kinnard, C.: Calving of Fuerza Aérea Glacier (Greenwich Island, Antarctica) observed
10  with terrestrial laser scanning and continuous video monitoring, J. Glaciol., 62(235), 835-846. doi:10.1017/jog.2016.72, 2016.

15

20

---

## Author Response (AR1)

We appreciate the insightful review and constructive comments from Andreas Köhler and the Anonymous Referee. The paper has been improved after addressing the concerns (see answers). Below we provide a marked-up manuscript and supplement.

Major changes are:

1. One new section in the main manuscript concerning long-term acoustic monitoring of calving fluxes.

2. Changed equations (11) and (12).

10  3. New figures and sections in the supplement.

[revised manuscript text omitted]
 energy computed for A1 and A2. The correlation decreases and the scatter increases when acoustic data from location A2 are used instead of A1 (see also Tab. 1 in the next section). Moreover, the regression line of the power-law fit is also different (Fig. S8). There are specific reasons behind this discrepancy.

Firstly, the signal-to-noise ratio for acoustic recordings taken at location A2 is lower than at A1. The receiver at A2 was more exposed to the contribution from sound sources located outside the glacial bay of Hansbreen (see Fig. 1 in the main manuscript). In addition, the mooring was located in shallow water (~22 m) so that a very low-frequency sound (< 50 Hz) propagates directly to the bottom, instead of being efficiently transmitted through the waveguide (e.g. Jensen et al., 2011). This results, for example, in a drop in signal-to-noise ratio when the frequency decreases from 50 to 10 Hz (Fig. 6f in the main manuscript). The shallow water damping of calving noise energy can be explored further. Figure S9 presents the correlation between the impact energy and uncorrected calving noise energy as a function of the upper frequency limit applied for the acoustic analysis. The correlation for location A2 is higher as frequency increases and the signal-to-noise ratio improves.

Secondly, spatial dependency of the thermohaline structure along the longer propagation path to A2 is expected to be significant and hard to simulate (see Fig. 2 in the main manuscript). The propagation model used in this study does not account for range-varying thermohaline structure.

[Figure]

Figure S8: Relationship between the block-water impact energy and underwater acoustic emission below 100 Hz for locations A1 (blue) and A2 (red).

[Figure]

Figure S9: Correlation between $E_{\mathrm{imp}}$ and $E_{\mathrm{ac,obs}}$ as a function of the cutoff frequency of the law-pass filter applied for acoustic recordings taken at locations A1 (blue) and A2 (red).

**5. Correlation between the log-transformed variables**

Table 1 presents correlation coefficients between log-transformed variables. The correlation between impact energy and noise energy for location A1 increases from 0.71 to 0.76 with the inclusion of transmission losses. However, it does not improve correlation for location A2. If the signal duration, $\Delta t$, is used instead of total energy of the signal, the correlation drops to 0.61. The duration of the received calving signal should not depend greatly on the propagation conditions in the bay, if the signal-to noise ratio is high enough. Therefore, it might be possible to use $\Delta t$ instead of the corrected noise energy to estimate iceberg calving fluxes (see best-fit line in Fig. S9), as an alternative when no information is given on bathymetry and sound speed profiles.

Table 1 Correlation coefficients between log-transformed variables. The duration of the calving signal, $\Delta t$, is the same for both locations: A1 and A2.

| | $E_{\mathrm{imp}}$ | $\Delta t$ | $E_{\mathrm{ac, obs}}$ ( A1 ) | $E_{\mathrm{ac, imp}}$ ( A1 ) | $E_{\mathrm{ac, obs}}$ ( A2 ) |
|---|---|---|---|---|---|
| $E_{\mathrm{ac, imp}}$ ( A2 ) | 0.30 | 0.15 | 0.64 | 0.53 | 1 |
| $E_{\mathrm{ac, obs}}$ ( A2 ) | 0.31 | 0.15 | 0.64 | 0.52 | |
| $E_{\mathrm{ac, imp}}$ ( A1 ) | 0.76 | 0.39 | 0.97 | | |
| $E_{\mathrm{ac, obs}}$ ( A1 ) | 0.71 | 0.34 | | | |
| $\Delta t$ | 0.61 | | | | |

[Figure]

Figure S10: Relationship between the block-water impact energy and duration of the calving signal.

---

## Author Response (AR2)

We very much appreciate the final comments from the handling editor Dr. Evgeny Podolskiy. The paper has been improved after addressing these concerns.

The main changes are:

1. New sentence on the sound absorption in section 3.4.1: *The absorption of sound in seawater is negligible for the low frequencies considered here (e.g. Ainslie and McColm, 1998).*

2. Updated caption to Fig. S4 in SI.

3. New references:

Ainslie, M. A., and McColm, J. G.: A simplified formula for viscous and chemical absorption in sea water, J. Acoust. Soc. Am., 103(3), 1671-1672, 1998.

Hatherton, T. and Evison, F. F.: A special mechanism for some Antarctic earthquakes, New Zeal. J. Geol. Geophys., 5:5, 864-873, https://doi.org/10.1080/00288306.1962.10417642, 1962.

Qamar, A.: Calving icebergs: a source of low-frequency seismic signals from Columbia Glacier, Alaska, J. Geophys. Res. Solid Earth, 93(B6), 6615–6623, https://doi.org/10.1029/JB093iB06p06615, 1988.

Richardson, J. P., Waite, G. P., FitzGerald, K. A., and Pennington, W. D.: Characteristics of seismic and acoustic signals produced by calving, Bering Glacier, Alaska, Geophysical Research Letters, 37(3), https://doi.org/10.1029/2009GL041113, 2010.

Walter, A., Lüthi, M. P., and Vieli, A.: Calving event size measurements and statistics of Eqip Sermia, Greenland, from terrestrial radar interferometry, The Cryosphere Discuss., https://doi.org/10.5194/tc-2019-102, in review, 2019.